# Ordered dephosphorylation initiated by the selective proteolysis of cyclin B drives mitotic exit

James Holder*, Shabaz Mohammed, Francis A Barr*

Department of Biochemistry, University of Oxford, Oxford, United Kingdom

**Abstract** APC/C-mediated proteolysis of cyclin B and securin promotes anaphase entry, inactivating CDK1 and permitting chromosome segregation, respectively. Reduction of CDK1 activity relieves inhibition of the CDK1-counteracting phosphatases PP1 and PP2A-B55, allowing wide-spread dephosphorylation of substrates. Meanwhile, continued APC/C activity promotes proteolysis of other mitotic regulators. Together, these activities orchestrate a complex series of events during mitotic exit. However, the relative importance of regulated proteolysis and dephosphorylation in dictating the order and timing of these events remains unclear. Using high temporal-resolution proteomics, we compare the relative extent of proteolysis and protein dephosphorylation. This reveals highly-selective rapid proteolysis of cyclin B, securin and geminin at the metaphase-anaphase transition, followed by slow proteolysis of other substrates. Dephosphorylation requires APC/C-dependent destruction of cyclin B and was resolved into PP1-dependent categories with unique sequence motifs. We conclude that dephosphorylation initiated by selective proteolysis of cyclin B drives the bulk of changes observed during mitotic exit.

**\*For correspondence:**
james.holder@path.ox.ac.uk (JH);
francis.barr@bioch.ox.ac.uk (FAB)

**Competing interests:** The authors declare that no competing interests exist.

## Introduction

Each cell cycle concludes with mitotic exit, wherein aligned chromosomes are segregated into two newly forming cells. This process has to be executed faithfully, since errors adversely impact the longevity and genomic stability of these new cells, thus contributing to a variety of pathologies, including cancer and neurological disorders (*Chunduri and Storchová, 2019*; *Gordon et al., 2012*). Mitotic exit is highly complex, involving wide-ranging changes in cell morphology, cytoskeletal architecture and genome packaging, alongside the reformation of membrane-bound organelles (*Lowe and Barr, 2007*). These dramatic events occur in a matter of minutes, necessitating a regulatory system that incorporates specific timing properties. Two modes of regulation dominate the current understanding of mitotic exit: ubiquitin-dependent proteolysis and protein dephosphorylation, mediated by the anaphase-promoting complex/cyclosome (APC/C) and phospho-protein phosphatases (PPP), respectively (*Holder et al., 2019*; *Nilsson, 2019*; *Sivakumar and Gorbsky, 2015*; *Watson et al., 2019*). Together these regulatory mechanisms co-ordinate the destruction of a cohort of proteins active during mitosis with global dephosphorylation of cyclin-dependent kinase 1 (CDK1) targets and local dephosphorylation of Polo and Aurora kinase targets. However, the relative contributions of these two regulatory branches to the successful completion of mitotic exit remains incompletely understood. This is due in part to the technical challenge of tracking both protein abundance and phosphorylation with sufficient time-resolution.

PP1 and PP2A-B55, both members of the PPP family, are thought to be the major phosphatases responsible for dephosphorylation during mitotic exit in human cells (*Capalbo et al., 2019*; *Castilho et al., 2009*; *Cundell et al., 2013*; *Cundell et al., 2016*; *Holder et al., 2019*; *McCloy et al., 2015*; *Mochida et al., 2009*; *Rodrigues et al., 2015*; *Schmitz et al., 2010*; *Trinkle-Mulcahy et al., 2006*; *Vagnarelli et al., 2011*; *Wu et al., 2009*). Consequently, PP1 and PP2A-B55

**eLife digest** New cells are made when a single cell duplicates its DNA and divides into two cells, distributing the DNA equally between them, in a process called mitosis. The splitting of the two copies of DNA happens through a series of controlled events known as mitotic exit. Previous research has suggested that mitotic exit relies on both the destruction of specific proteins and the removal of tags called phosphate groups from other proteins. Phosphate groups modify how proteins behave and their removal can trigger changes in a protein's activity.

Although protein destruction and phosphate group removal were known to be important to mitotic exit, it was not understood how they are coordinated in the cell to ensure the correct order of events. Holder et al. have used a technique called mass spectrometry to monitor the level of thousands of proteins, and any tags attached to them, during mitotic exit in human cells grown in the laboratory.

The experiments revealed that the destruction of a single protein, known as cyclin B, plays a major role in triggering subsequent events. The removal of cyclin B activates enzymes known as phosphatases, which remove phosphate groups from proteins. Phosphatases then act on a wide range of proteins in a specific order that depends on the environment surrounding the phosphate group. This 'chain' of phosphatase activity determines the order of events during mitotic exit.

The findings of Holder et al. contribute to the basic understanding of how mitotic exit works. Errors in the process can affect the stability of a cell's genome, contributing to diseases such as cancer. In the future, this may help to identify what goes wrong in these cases and potential avenues for developing treatments.

are inhibited by the cyclin B-dependent kinase 1 (CDK1-cyclin B) following mitotic entry, allowing mitotic phosphorylation to accumulate. PP1 is directly inhibited by CDK1-cyclin B through phosphorylation in the C-terminal tail at PP1-T320 (*Dohadwala et al., 1994*; *Yamano et al., 1994*). Inhibition of PP2A-B55 is indirect and mediated by the B55-ENSA-Greatwall (BEG) pathway (*Castilho et al., 2009*; *Cundell et al., 2013*; *Gharbi-Ayachi et al., 2010*; *Mochida et al., 2010*). ENSA itself is only slowly dephosphorylated by PP2A-B55, leading to the characteristic delay in dephosphorylation of PP2A-B55 substrates following cyclin B destruction (*Cundell et al., 2013*; *Williams et al., 2014*). Importantly, PP1 activity is subject to further regulation through the action of PP1-interacting proteins, which impact substrate selectivity and in some cases directly inhibit PP1 activity (*Bollen et al., 2010*). Many of these interactors are themselves subject to phospho-regulation, such as Inhibitor-1 and Repo-man, and their PP1-dependent dephosphorylation is required for full activation of PP1 (*Qian et al., 2015*; *Wu et al., 2018*; *Wu et al., 2009*). Therefore, destruction of cyclin B triggers inactivation of CDK1 and relieves the inhibition of the opposing phosphatases, PP1 and PP2A-B55, resulting in global protein-dephosphorylation.

PP1 and PP2A share highly similar catalytic subunits and individually do not demonstrate different substrate specificities, necessitating the presence of regulatory subunits to target their activities appropriately (*Agostinis et al., 1992*; *Imaoka et al., 1983*; *Mumby et al., 1987*). PP1 catalytic subunits, PPP1CA/B/C, can bind hundreds of regulatory subunits (*Bollen et al., 2010*). In contrast, B55 regulatory subunits interact with both PPP2CA/B catalytic and a scaffold subunit to form PP2A-B55 holoenzymes (*Xing et al., 2006*; *Xu et al., 2006*). The phospho-site preferences of PPPs have previously been shown to impact the order of events during mitotic exit. An acidic patch on the surface of the B55 subunit confers a preference to PP2A-B55 holoenzymes for phospho-sites flanked by a bi-partite basic motif (*Cundell et al., 2016*). Substrates with the highest rates of B55-dependent dephosphorylation are the most basic, typically the proteins involved in spindle morphology changes such as PRC1 and TPX2. Conversely, the slowest dephosphorylated substrates are less basic and include nuclear pore and envelope proteins such as nuclear pore complex protein 153 (NUP153). This helps ensure that nuclear envelope reformation and import occur downstream of chromosome segregation and central spindle formation (*Cundell et al., 2016*). Furthermore, phospho-threonine is more readily dephosphorylated by PP2A-B55 than phospho-serine, impacting spindle assembly checkpoint responsiveness and APC/C co-factor selection, and therefore substrate specificity (*Hein et al., 2017*). A proteomic study conducted in HeLa cells suggested that these phospho-site

preferences apply to the bulk of phosphatases active in mitotic exit, not just to PP2A-B55 (*McCloy et al., 2015*). Nevertheless, this analysis was limited to one timepoint during early mitotic exit under conditions of proteasome inhibition. A further analysis of mitotic exit in budding yeast found that early dephosphorylations were more basic than later dephosphorylations, correlating with the kinase motifs of CDK and PLK1, respectively (*Touati et al., 2018*). However, it remains to be seen if these results are directly applicable to mammalian cells since Cdc14 is the primary mitotic exit phosphatase in budding yeast (*Stegmeier and Amon, 2004*) and current evidence suggests that this role is not conserved in human cells (*Berdougo et al., 2008*; *Mocciaro et al., 2010*). Furthermore, Cdc14 regulation through the FEAR and MEN pathways is highly divergent from the CDK1-mediated inhibition of PP1 and PP2A-B55, therefore timing properties are not directly comparable between yeast and human cells.

CDK1-inactivation and the loss of chromosome cohesion are required to initiate mitotic exit. This is achieved through the APC/C-mediated destruction of cyclin B and securin, respectively (*Funabiki et al., 1996*; *King et al., 1995*; *Sudakin et al., 1995*). The APC/C is a multi-subunit E3-ubiquitin ligase (*Hutchins et al., 2010*; *Zachariae et al., 1996*) which requires the presence of a coactivator, either CDC20 or CDH1, to ubiquitinate substrates (*Kramer et al., 1998*; *Visintin et al., 1997*). Both CDC20 and CDH1 make similar contacts with the APC/C, however only CDC20 binding is promoted by mitotic APC/C phosphorylation (*Kraft et al., 2003*; *Zhang et al., 2016*). APC/C coactivator complexes recognise substrates through degron motifs, typically the D-box, whose presence is necessary and sufficient for APC/C-mediated ubiquitination of particular substrates (*Glotzer et al., 1991*; *King et al., 1996*). Alternatively, proteins can be targeted through the KEN box, either alone or in combination with a D-box (*Burton and Solomon, 2001*; *Passmore et al., 2003*; *Pfleger and Kirschner, 2000*). Alternative degron motifs have also been identified such as the ABBA motif, found in cyclin A, BUBR1 and BUB1, which is recognised by human CDC20 (*Davey and Morgan, 2016*; *Di Fiore et al., 2015*; *Di Fiore and Pines, 2010*; *Lu et al., 2014*). Importantly, mitotic kinases such as Aurora A, Aurora B and PLK1 are degraded later than cyclin B during mitotic exit (*Floyd et al., 2008*; *Lindon and Pines, 2004*). This creates a cell state with low CDK and high Aurora and PLK1 activities fundamentally different to that found earlier in mitosis or during G2. This asymmetry in kinase activity surrounding mitosis is central to directionality during mitotic exit (*Holder et al., 2019*). Furthermore, sustained activity of Aurora B and PLK1 is important for local, spatial regulation of mitotic exit including the successful completion cytokinesis and abscission (*Afonso et al., 2014*; *Bastos and Barr, 2010*; *Bastos et al., 2014*; *Nunes Bastos et al., 2013*; *Steigemann et al., 2009*). Interestingly, recent work has shown that small pools of CDK1-cyclin B activity persist into anaphase contributing to this local regulation (*Afonso et al., 2019*; *Mathieu et al., 2013*).

A further complication when analysing mitotic exit is the potential contribution of protein synthesis. Mitotic factors, including cyclin B, Aurora and Polo kinases and mitotic spindle factors, are predominantly synthesised in G2 (*Whitfield et al., 2002*). APC/C activity is then required to ensure that these proteins are degraded in a timely manner during mitotic exit. In a 2011 review, 55 human proteins were identified as APC/C substrates (*Meyer and Rape, 2011*). During mitosis, the spindle assembly checkpoint promotes assembly of the mitotic checkpoint complex (MCC) from MAD2, BUB3, BUBR1 and CDC20 at unattached kinetochores to inhibit APC/C activity against anaphase substrates (*Fang, 2002*; *Tipton et al., 2011*). CDC20 is unique amongst the MCC proteins in that it is simultaneously a coactivator and inhibitor of the APC/C (*Alfieri et al., 2016*; *Izawa and Pines, 2015*; *Yamaguchi et al., 2016*). Once the checkpoint is satisfied and the MCC is disassembled APC/C$^{CDC20}$ becomes active, leading to ubiquitination and proteasomal destruction of many substrates. It has been proposed that continual synthesis and destruction of CDC20 is required for rapid APC/C$^{CDC20}$ activation upon checkpoint satisfaction (*Foe et al., 2011*; *Foster and Morgan, 2012*; *Nilsson et al., 2008*; *Varetti et al., 2011*; *Wang et al., 2017*). This destruction of CDC20 is dependent on a functional spindle checkpoint and is mediated by APC15 (*Mansfeld et al., 2011*; *Uzunova et al., 2012*). Furthermore, the ubiquitination of CDC20 has itself been suggested to promote MCC disassembly (*Kallio et al., 1998*; *Reddy et al., 2007*).

The MCC ensures ordered cyclin proteolysis during mitosis, permitting APC/C activity against cyclin A throughout mitosis whilst preventing ubiquitination of cyclin B (*Di Fiore et al., 2015*; *Geley et al., 2001*; *Jacobs et al., 2001*; *Wolthuis et al., 2008*; *Zhang et al., 2019*). It has also been proposed that ordered proteolysis contributes to a proper mitotic exit (*Lindon and Pines, 2004*).

The APC/C^CDH1 complex can form as cyclin B is degraded and CDK1 activity falls in an APC/C^CDC20-mediated manner (*Kramer et al., 2000*; *Kotani et al., 1999*; *Visintin et al., 1997*). APC/C^CDH1 can then target CDC20 for destruction (*Pfleger and Kirschner, 2000*). This rapid change in APC/C coactivator alters substrate specificity, contributing to ordered proteolysis during mitotic exit. Once active, APC/C^CDH1 can target cytokinesis regulators, such as Aurora kinases, TPX2, PLK1, anillin and cytoskeletal associated protein 2 (CKAP2) for destruction (*Floyd et al., 2008*; *Lindon and Pines, 2004*; *Seki and Fang, 2007*; *Stewart and Fang, 2005*; *Taylor and Peters, 2008*; *Zhao and Fang, 2005*). It has been suggested that PLK1 is degraded prior to Aurora A (*Lindon and Pines, 2004*), to ensure proper spindle dynamics and organisation during anaphase (*Floyd et al., 2008*). Interestingly, few proteins, beside Aurora kinases, have been shown to be completely dependent on APC/C^CDH1 for their destruction (*Floyd et al., 2008*). Indeed, recent work has shown that while Aurora A levels, but not activity, are stabilized in the absence of CDH1, the Aurora A targeting protein and activator TPX2 is degraded normally (*Abdelbaki et al., 2020*). Furthermore, depletion of CDH1 stabilises levels of anillin and CKAP2 but does not affect the kinetics of mitotic exit or cytokinesis (*García-Higuera et al., 2008*; *Seki and Fang, 2007*), raising questions regarding the importance of switching coactivators during mitotic exit in human cells. Global proteomic analysis of meiotic exit in *Xenopus* eggs, which lack CDH1, showed that proteolysis was limited to a few key rapidly degraded targets such as cyclin B1, B2 and securin and that dephosphorylation proceeded normally in this system (*Presler et al., 2017*). Additionally, proteasome inhibition, using MG-132, did not alter the rate of dephosphorylation of PPP1CA-pT320 or PRC1-pT481 in HeLa cells following CDK inhibition (*Cundell et al., 2013*). These latter results suggest that APC/C-dependent proteolysis fulfils a more limited role than previously thought and may be more important for resetting cell status in G1 rather than implicitly impacting the order and timing of events during mitotic exit.

Here, we use high-resolution mass spectrometry to show that proteolysis of the crucial APC/C substrate, cyclin B1 (CCNB1), triggers a graded series of dephosphorylations whose rate is encoded into the system at the level of individual phospho-sites. We find that one important general feature of this system is that PPPs will more readily dephosphorylate phospho-threonine in a basic environment, both at the level of an individual phosphatase, PP1, or the system as a whole. Significantly, these dephosphorylations proceed normally in the absence of APC/C activity or protein translation following CDK inhibition, demonstrating that ordered proteolysis and protein synthesis alone are insufficient to co-ordinate the events of mitotic exit. Instead we suggest that the main orchestration of mitotic exit events is mediated by regulated dephosphorylation.

## Results

### Ordered dephosphorylation, not proteolysis, is the dominant mode of anaphase regulation

To gain a more detailed quantitative understanding of mitotic exit it was important to be able to follow changes in both protein abundance and phosphorylation state in vivo with high temporal resolution. To achieve this, HeLa cells were synchronised in mitotic prometaphase by washout from nocodazole arrest for 25 min to allow spindle formation and chromosome alignment. To promote entry into anaphase with a high degree of synchrony, CDK1 was then inhibited using flavopiridol and samples taken every 30 s until 10 min and at increasing intervals up to 60 min (*Figure 1A*). Markers of mitosis and anaphase were followed by western blot as cells progressed through mitotic exit (*Figure 1A*). The use of chemical CDK inhibition to trigger mitotic exit uncouples CDK1 activity from the level of CCNB1, allowing cells to exit mitosis prematurely prior to CCNB1 proteolysis. Consequently, some dephosphorylations take place earlier than under physiological conditions (*Cundell et al., 2013*; *Cundell et al., 2016*). This system collapses the characteristic delay seen before dephosphorylation of PP2A-B55 sites, such as PRC1-pT481 (*Figure 1A and B*; *Cundell et al., 2013*). Following CDK inhibition, both PPP1CA-pT320, an indicator of PP1 activity and PRC1-pT481 are rapidly dephosphorylated with near complete loss of signal by 10 min (*Figure 1A–C*). This indicates that the inhibition of PP1 and PP2A-B55, respectively, has been relieved. Additionally, the anaphase-specific PLK1 phosphorylation of PRC1-pT602 began to accumulate precociously from 2.5 min, as the PRC1-pT481 signal decreased (*Figure 1A and B*). Finally, APC/C-dependent proteolysis was observed as highlighted by the decreasing levels of CCNB1 and securin (*Figure 1A and D*).

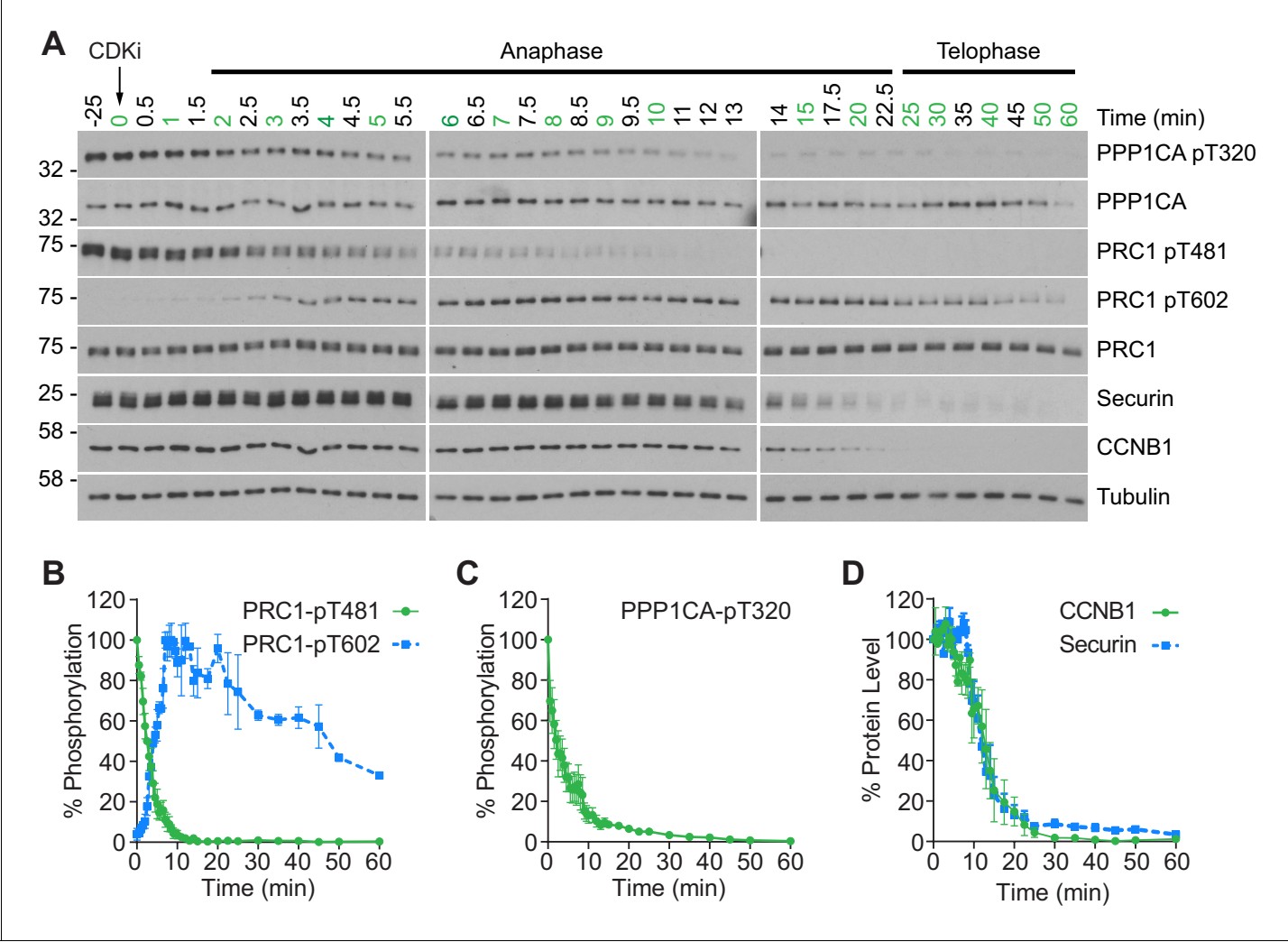

**Figure 1.** CDK inhibition promotes orderly mitotic exit. (A) Mitotically arrested cells were washed out from nocodazole, at 37°C 5% $CO_2$, and the CDK inhibitor (CDKi) flavopiridol was added at 0 min. Samples were taken at the indicated timepoints, with those used for subsequent mass spectrometry analysis highlighted in green. Densitometric quantification of repeat experiments, as in (A) (Mean ± SEM) is shown for (B) PRC1-pT481 (n = 3), pT602 (n = 2), (C) PPP1CA-pT320 (n = 3), and for (D) CCNB1 and securin protein level (n = 3).

Together these markers indicate that the cells had undergone mitotic exit and progressed into anaphase in an orderly fashion.

Four repeats of these time course samples were processed for mass spectrometry using quantitative di-methyl labelling followed by phospho-peptide enrichment (*Boersema et al., 2009*). This resulted in the high confidence identification of 4813 proteins and 18,675 phosphorylation sites (*Figure 2—source data 1* and *2*). Global trends in the data were identified through a clustering analysis, grouping proteolysis or dephosphorylation events into four categories: early, intermediate, late or stable. Following processing, 2165 proteins and 4632 phospho-sites were clustered (*Figure 2A–D*). Behaviour of the reference substrates CCNB1, PRC1 and PRC1-pT481 show that the proteomic data accurately recapitulates that obtained by western blot (*Figure 2E–G*).

Strikingly, given the number of APC/C substrates and proteins synthesised for mitosis, the vast majority of the total proteome was stable during the time window observed (*Figure 2A*). In fact, the only protein seen to be degraded rapidly using this global proteomic approach was CCNB1 (*Figure 2E*). Detection of securin was inconsistent due to the frequency and distribution of basic residues cleaved during a tryptic digest. A small number of proteins, such as PLK1, were degraded slowly during the time window analysed while the only proteins which demonstrated an intermediate

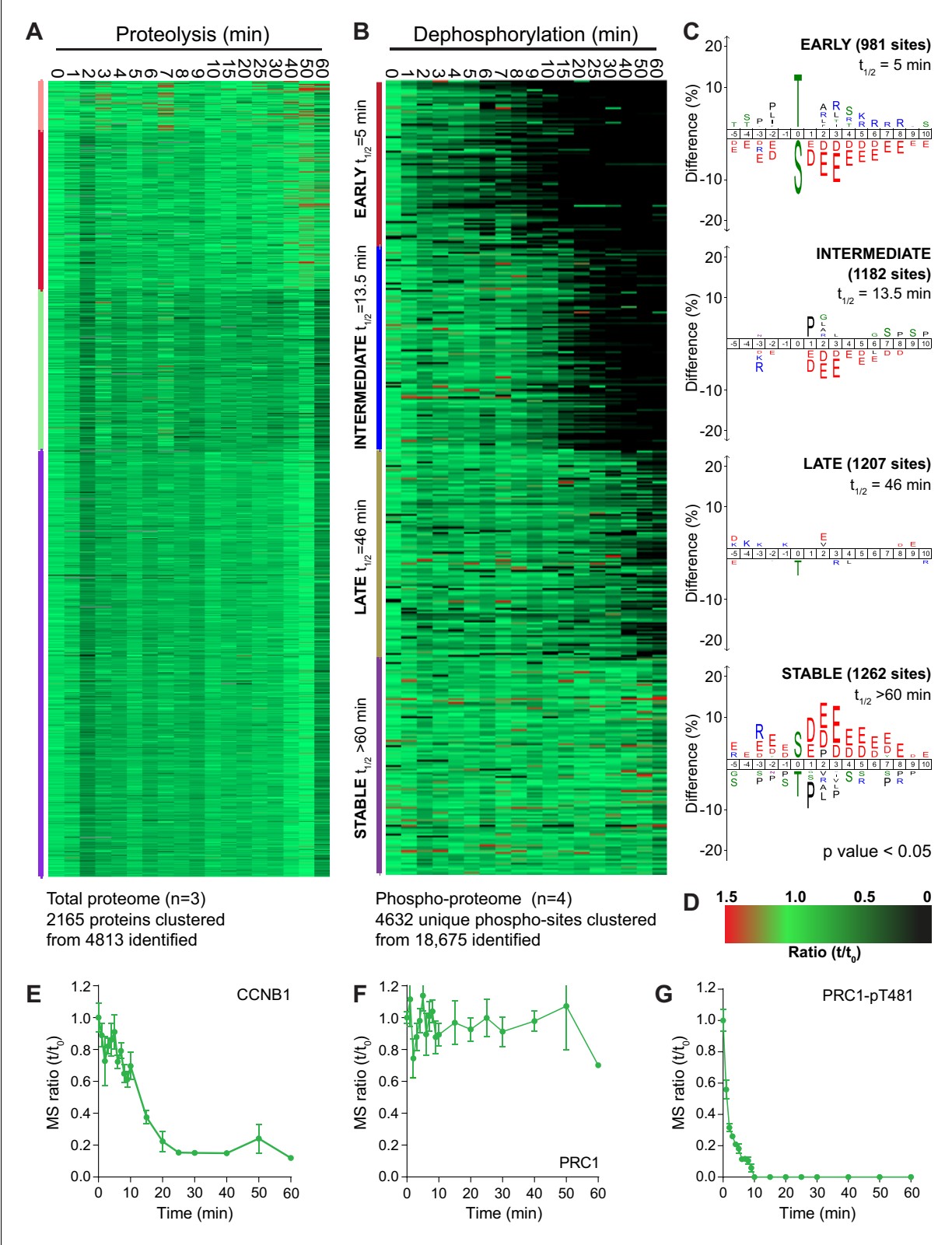

**Figure 2.** An ordered cascade of dephosphorylation during mitotic exit. HeLa cells were arrested in mitosis and mitotic exit triggered as previously described in *Figure 1*. Heatmap of (**A**) protein groups (*Figure 2—source data 1*) or (**B**) phospho-sites (*Figure 2—source data 2*) clustered based on trends over time. The coloured bar adjacent to each heatmap indicates the limits of each cluster, with each coloured section corresponding to one cluster. Time is shown in minutes. (**C**) Sequence logo analysis showing the relative enrichment of residues in the adjacent cluster from (**B**). The number

*Figure 2 continued on next page*

*Figure 2 continued*

of phospho-sites and average half-life for each cluster is indicated. (**D**) Key linking heatmap colour with observed ratio. Quantitative ratio of (**E-F**) CCNB1 and PRC1 protein level (n = 3) or (**G**) PRC1-pT481 phosphorylation (Mean ± SEM, n = 4).

The online version of this article includes the following source data and figure supplement(s) for figure 2:

**Source data 1.** Total proteome data and clustering analysis of mitotic exit triggered using CDK inhibtion.
**Source data 2.** Phospho-proteome data and clustering analysis of mitotic exit triggered using CDK inhibtion.
**Source data 3.** Total and phospho-proteomic data and clustering analysis of mitotic exit triggered using MPS1 inhibtion.
**Figure supplement 1.** An ordered cascade of dephosphorylation in anaphase A and B.

rate of proteolysis were Aurora A and its coactivator TPX2. To corroborate these findings previously published proteomic data, generated by the lab using MPS1-inhibition to trigger mitotic exit rather than CDK-inhibition (*Hayward et al., 2019*), was also inspected for changes at the protein level and subjected to the same clustering analysis as above (*Figure 2—figure supplement 1* and *Figure 2—source data 3*).

In this data set CCNB1 destruction followed more typical delayed sigmoidal kinetics during mitotic exit (*Figure 2—figure supplement 1E*), whilst the proteome as a whole remains stable following MPS1 inhibition, exemplified by PRC1 (*Figure 2—figure supplement 1A and F*). This means that under conditions of either normal (MPS1i) or collapsed mitotic exit (CDKi) the detectable fraction of the proteome was stable for the window observed. This fitted with expectations, as only the destruction kinetics should change between the two conditions, rather than the entire substrate specificity of the APC/C. Of all the detected proteins only CCNB1 and securin (western blot only) exhibited rapid enough destruction to impact the events of anaphase. Importantly, in unperturbed cells, these destruction events occur prior to the onset of anaphase. These data would question the role of ordered proteolysis in the regulation of mitotic exit, beyond triggering anaphase entry.

By contrast, when similarly analysed, dephosphorylations clustered neatly into four groups; early, intermediate, late and stable (*Figure 2B*). Sequence-logo analysis of each cluster highlights a clear trend in phospho-site sequence characteristics. Early dephosphorylations, with an average half-life of 5 min, were strongly enriched for phospho-threonine over phospho-serine (*Figure 2C*, early). Surrounding the phospho-site, a mild enrichment of basic residues and a strong de-enrichment of acidic residues could be observed (*Figure 2C*, early). Intermediate dephosphorylations, with a typical half-life of 13.5 min, were also de-enriched of acidic residues, albeit to a lesser extent than the early cluster (*Figure 2C*, intermediate). Late dephosphorylations with an average half-life of 46 min showed little dominant positive or negative selections, save for a mild selection against phospho-threonine (*Figure 2C*, late). Stable phosphorylation, defined as sites with a half-life greater than 60 min, had a very strong enrichment of acidic residues surrounding the phospho-site and some enrichment for phospho-serine (*Figure 2C*, stable). Phospho-threonine and basic residues were modestly selected against within the stable population. This analysis shows a gradual and continuous trend consistent with the published literature whereby the more acidic a phosphorylation site, the slower its dephosphorylation (*McCloy et al., 2015*; *Touati et al., 2018*). Additionally, all else being equal, a phospho-threonine is more likely to be dephosphorylated rapidly than a phospho-serine (*Cundell et al., 2016*; *Hein et al., 2017*).

A clustering analysis of data generated following MPS1 inhibition yielded the same trends in sequence characteristics seen after CDK1 inhibition (*Figure 2—figure supplement 1B and C*). Only two clusters of dephosphorylation could be observed in this analysis (*Figure 2—figure supplement 1B*, early and intermediate). Since MPS1 inhibition causes a gradual decline in CDK1 activity, via CCNB1 destruction, many dephosphorylations will be slower under these conditions, as demonstrated by the profiles of PPP1CA-pT320 and PRC1-pT481 (*Figure 2—figure supplement 1G*). Two stable clusters of phospho-sites could be observed, one of which displayed the sequence traits expected (*Figure 2—figure supplement 1C*, stable 1). The second stable cluster contained noisier data, as seen by an increase in randomly scattered black marks, representative of missing values (*Figure 2—figure supplement 1C*, stable 2). This led to a logo without any apparent preferences. Taken together these data suggest that it is ordered dephosphorylation and not proteolysis which is responsible for anaphase regulation following mitosis. Furthermore, the rate of these

dephosphorylations is encoded into the system through the amino acid sequence characteristics surrounding each phospho-site.

## The majority of APC/C-mediated proteolysis occurs downstream of anaphase

Although over 2000 proteins were reliably quantified across the time course, it remained possible that lower abundance proteins were being degraded but could not be detected due to instrument limitations. Therefore, high pH fractionation was used to increase the depth of the total-proteome, using samples taken every 10 min across two repeats of the time course. With this approach over 5600 proteins were identified, with 3748 being reliably quantitated (*Figure 3—source data 1*). The profile of CCNB1 in the fractionated samples (*Figure 3—figure supplement 1C*) was consistent with western blotting and our initial proteomic approach (*Figures 1D* and *2E*, respectively). The greater sensitivity afforded by high pH fractionation enabled the detection of lower abundance, lysine-rich proteins such as CDC20 and the DNA replication licensing regulator geminin (GMNN) (*Figure 3—figure supplement 1D and E*). Importantly, clustering analysis showed that the majority of the proteome was stable even at this greater depth (*Figure 3—figure supplement 1A*). Additionally, ~50% CDC20 was still present one hour after CDK inhibition. This would suggest that the switch from CDC20- to CDH1- dependent destruction may not be as clear as was once thought. The representation of APC/C substrates and their half-lives, from within this dataset, was then assessed. Firstly, of the 55 human proteins identified as substrates of the APC/C (*Meyer and Rape, 2011*), 34 were identified and quantified here (*Figure 3A*). Additionally, of the 210 genes shown to demonstrate G2/M or M/G1 expression patterns (*Whitfield et al., 2002*), 120 are identified in the fractionated data set (*Figure 3A*). To ensure unbiased classification of whether a protein was degraded during the experimental window, a threshold value was selected. Since the events of anaphase happen on a timescale of minutes this threshold was set to a half-life of 60 min. When analysing both reference lists, only

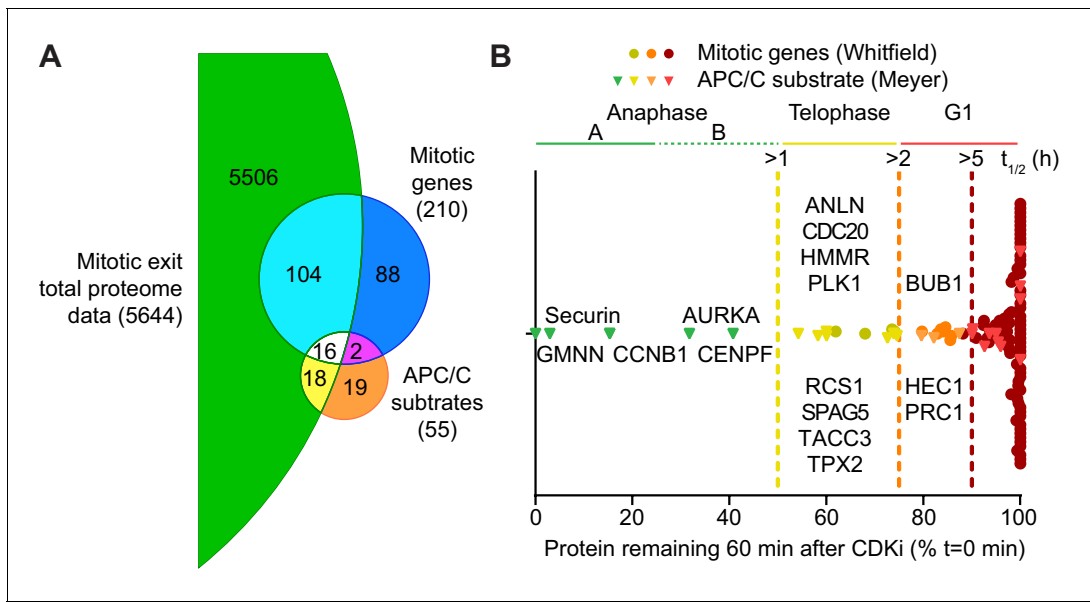

**Figure 3.** Selective proteolysis of cyclin B, geminin and securin in anaphase A. (**A**) Venn diagram comparison of the proteins identified here by mass spectrometry with mitotic genes transcribed in G2 of the cell cycle and validated APC/C substrates. These reference lists were obtained from *Whitfield et al., 2002* and *Meyer and Rape, 2011*, respectively. Numbers indicate how many identifications are in each region. (**B**) Proteins identified by mass spectrometry plotted according to the percentage of protein remaining 60 min after CDK inhibition (*Figure 3—source data 1*). Corresponding western blot samples were used for the analysis of securin levels following CDK inhibition. Vertical lines indicate various set half-lives for degrading proteins. The corresponding cell cycle stage is indicated above the graph. Proteins found in both the Whitfield and Meyer papers were classified as APC/C substrates.

The online version of this article includes the following source data and figure supplement(s) for figure 3:

**Source data 1.** Fractionated total proteome data and clustering analysis of mitotic exit triggered using CDK inhibtion.
**Figure supplement 1.** Clustering analysis of fractionated anaphase proteomes.

five proteins meet the criterion of APC/C-mediated anaphase destruction: Aurora A, CENPF, CCNB1, GMNN and securin (*Figure 3B*, green triangles). Indeed, CCNB1, GMNN and securin are degraded prior to anaphase onset in unperturbed cells, while proteolysis of Aurora A and CENPF occurred too slowly to impact Anaphase A. If this half-life threshold was relaxed to 2 hr to include proteolytic events occurring in telophase or early G1, an additional 11 proteins would meet the destruction criteria. (*Figure 3B*, left of orange line). With the exception of GMNN and securin, these 16 degraded proteins fall into two categories; either spindle assembly checkpoint, kinetochore and centromere proteins or mitotic spindle factors. A small number of proteins such as PRC1, HEC1 and BUB1 had half-lives between 2–5 hr placing their proteolysis in the subsequent G1 phase (*Figure 3B*, left of orange line). The remaining 89 proteins all had half-lives greater than 5 hr. This additional analysis supports the conclusion that only a small portion of the mitotic proteome is subject to destruction and very few of these proteolytic events could impact the order of events in anaphase.

## Protein synthesis is not required for progression through the metaphase-anaphase transition

It has been proposed that continued synthesis of key mitotic proteins including CDC20 and CCNB1 is essential for sustained spindle checkpoint function (*Mena et al., 2010*; *Nilsson et al., 2008*; *Varetti et al., 2011*). Protein synthesis could therefore impact mitotic exit, potentially masking APC/C activity towards many substrates. To assess the contribution of protein synthesis to mitosis and mitotic exit, CRISPR-tagged CCNB1-GFP cells (*Alfonso-Pérez et al., 2019*) were followed by live cell microscopy. Cycloheximide (CHX), an inhibitor of protein synthesis, or DMSO were added just prior to cells entering mitosis. In control cells the level of CCNB1 continued to climb throughout the experiment until cells entered mitosis (*Figure 4A and B*, upper panels). CCNB1 levels peaked and briefly plateaued before rapidly decreasing. These features indicated that the cells had entered mitosis, satisfied the checkpoint and entered anaphase. Importantly, the presence of a plateau indicates that levels of CCNB1 do not continue to rise throughout mitosis. Upon addition of CHX, levels of CCNB1 ceased to increase, confirming the activity of the drug (*Figure 4A and B*, lower panels). As such, many of the cells imaged did not have the necessary levels of CCNB1 to enter mitosis. In these cells the CCNB1 fluorescence only gradually declined, more indicative of bleaching than of active destruction (*Figure 4B*, lower panel, G2 arrest). Cells which had accumulated adequate levels of CCNB1 before the addition of CHX entered mitosis (*Figure 4B*, lower panel, Mitotic), and spent a similar time in mitosis to the control cells (*Figure 4C*).

However, due to the number of CHX treated cells analysed the possibility of small differences being present cannot be excluded (*Figure 4C*). Importantly, unlike MAD2 depletion (*Michel et al., 2004*), cells treated with CHX did not collapse out of mitosis, suggesting that cells were able to support spindle checkpoint function and enter anaphase following destruction of CCNB1.

Checkpoint function was then interrogated directly under conditions of inhibited protein synthesis by treating these cells simultaneously with nocodazole and either DMSO or CHX. In control cells, CCNB1 fluorescence increased until mitotic entry followed by a small plateau, after which the fluorescence gradually decreased (*Figure 4—figure supplement 1A and B*, upper panels). Since the cells were arrested in mitosis during imaging (*Figure 4—figure supplement 1A*), this decrease in signal is again likely due to fluorophore bleaching rather than protein destruction. When treated with CHX and nocodazole, CCNB1 fluorescence began decreasing immediately following drug addition, at a similar rate to that seen in control cells (*Figure 4—figure supplement 1A and B*). Importantly, the cells in both groups were still arrested in mitosis 360 min after drug addition, meaning CCNB1 and CDC20 must still be sufficiently present to permit checkpoint function (*Figure 4—figure supplement 1A*, lower panel).

Changes in protein level were then followed by western blot in cells arrested in mitosis using either nocodazole or taxol treatment to activate the spindle checkpoint. BUBR1 and MAD2 were both stable for two hours following inhibition of protein synthesis (*Figure 4—figure supplement 1C and D*). During this period CCNB1 levels fell to 80% of their starting value, however the levels were barely different from the control during the first 60 min (*Figure 4—figure supplement 1C–E*). Across the two hours approximately 50% of CDC20 was destroyed (*Figure 4—figure supplement 1F*). This is inconsistent with the published high rates of protein synthesis and turnover during mitosis in mammalian cells (*Nilsson et al., 2008*; *Varetti et al., 2011*). However, those published results were

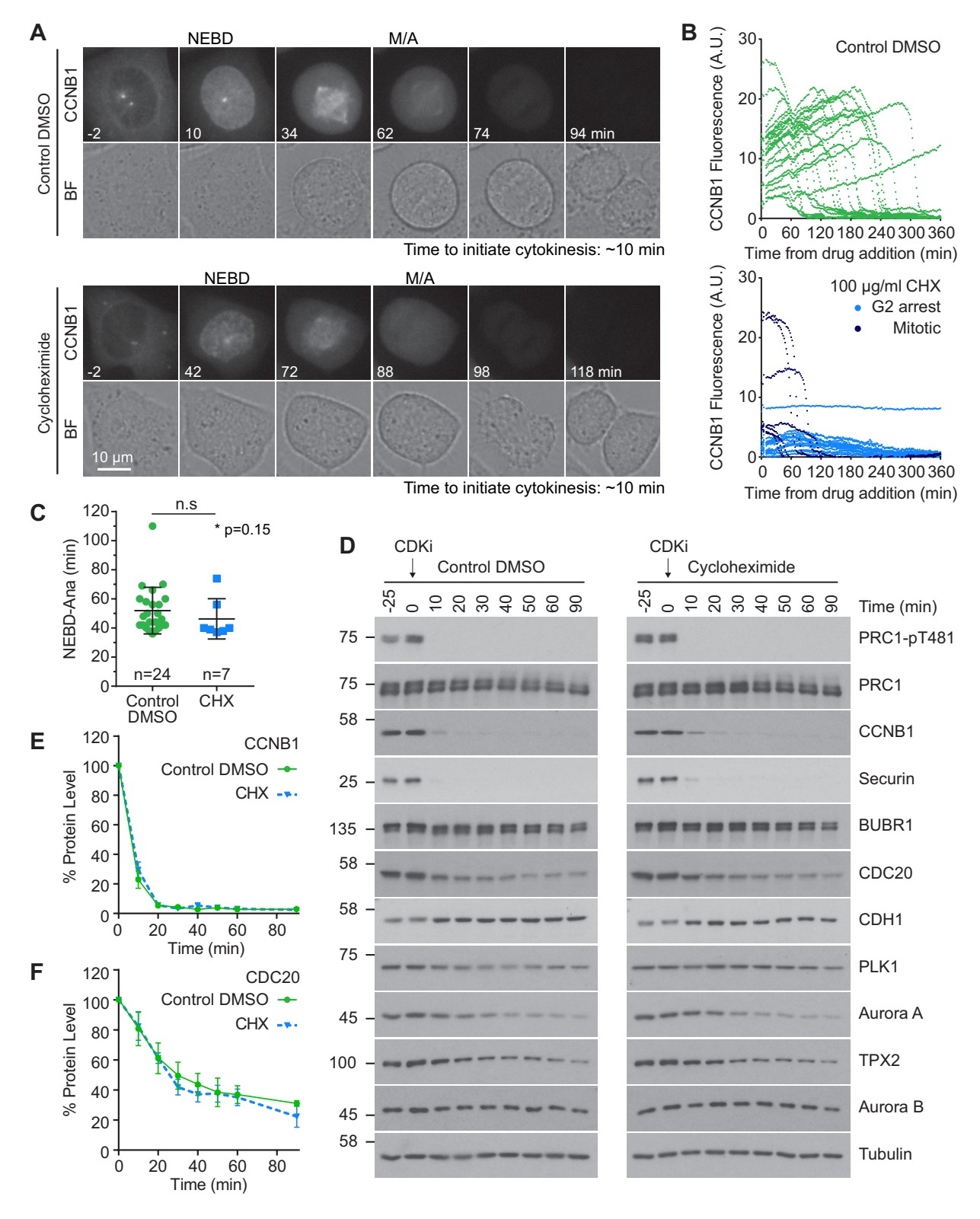

**Figure 4.** Protein synthesis does not impact mitotic progression nor account for the differences in the apparent rate of proteolysis during mitotic exit. (A) Live cell imaging of endogenously CRISPR-tagged CCNB1, following addition of either DMSO as a control or 100 μg/ml cycloheximide (CHX). (B) Fluorescence intensity plot of all cells within the field of view following drug addition (DMSO n = 24, CHX n = 21, from a single experiment). Times shown in (A–B) are relative to the point of drug addition. (C) Statistical analysis of the time cells from (A) spent in mitosis using a Mann-Whitney test, *
*Figure 4 continued on next page*

*Figure 4 continued*

denotes the calculated p-value (DMSO n = 24, CHX n = 7). Images shown are sum intensity projections, quantification was carried out on sum intensity projections. (D) Mitotically arrested HeLa cells were washed out from nocodazole and treated with either DMSO or 100 μg/ml CHX for 25 min prior to CDK inhibition with flavopiridol. Samples were harvested as indicated and analysed by western blot. (E–F) Densitometric quantification of protein level, from western blot data (D), for indicated protein (Mean ± SEM, n = 3).

The online version of this article includes the following figure supplement(s) for figure 4:

**Figure supplement 1.** Ongoing protein synthesis is not required to maintain spindle assembly checkpoint-dependent arrest during mitosis.

obtained using a CDC20 antibody directed against an epitope within the C-terminal 50 amino acids of the protein. This region is ubiquitinated in an APC15-depdendent manner under conditions of checkpoint activation (*Uzunova et al., 2012*), therefore the rapid destruction observed is likely a consequence of epitope masking (*Nilsson et al., 2008*; *Uzunova et al., 2012*). Importantly, the changes in protein level here were followed over a 120 min period, whereas a normal mitosis typically takes ~ 60 min. These results suggest that ongoing protein synthesis and turnover is not essential for spindle checkpoint-dependent arrest during mitosis or progression into anaphase.

We then asked if protein synthesis was required to complete anaphase A and initiate cytokinesis using time lapse imaging. In both control and CHX-treated cells, the mean time for furrow ingression following chromosome separation was ~ 9 min (*Figure 4—figure supplement 1F*). To provide further support for these results, mitotically arrested HeLa cells were pre-treated with CHX prior to the addition of CDK inhibitor. Destruction of proteins during mitotic exit was then followed by western blotting and was found to be unaffected by inhibition of protein synthesis (*Figure 4D*). This was confirmed by quantification of CDC20 and CCNB1 (*Figure 4E and F*). Furthermore, if continued protein synthesis was critical to a normal mitosis it would follow that a 25 min treatment with CHX, equivalent to roughly half a normal mitosis, should cause a measurable decrease in protein levels. However, no difference was observed when comparing the levels of different mitotic proteins in the −25 and 0 min timepoints in either control or synthesis-inhibited conditions (*Figure 4D*). Critically, dephosphorylation of PRC1-pT481 was unaffected by the addition of CHX (*Figure 4D*). These results suggest that protein synthesis is not required for normal cellular function during mitosis and anaphase. Furthermore, our results demonstrate that continued protein synthesis into anaphase is insufficient to explain the observed stability of the total proteome.

## Anaphase dephosphorylations are unperturbed by the absence of APC/C activity

The data show that beyond the initial destruction of CCNB1, GMNN and securin, protein level homeostasis is not responsible for regulating the events of anaphase. If regulation of anaphase is indeed dominated by phosphatase activity then substrate dephosphorylation should be unaffected by the absence of proteolytic activity (*Cundell et al., 2013*). Owing to the presence of many ubiquitin ligases within the cell, direct inhibition of the proteasome, using MG-132, is not specific enough to test this. Therefore, the APC/C inhibitors APC inhibitor (apcin) and pro-tosyl-L-arginine methyl ester (pro-TAME) were used. These have been shown to function synergistically, providing optimal results when used together (*Sackton et al., 2014*).

An initial low-resolution time course was carried out to assess dephosphorylation under conditions of APC/C inhibition. Cells were prepared as before, and pre-treated with either DMSO, APC/C inhibitors alone or APC/C and proteasome inhibitors (40 μM MG-132) prior to CDK inhibition. The functional consequences of these treatments are highlighted in *Figure 5A*. CCNB1 was completely degraded by 20 min in control conditions and was fully stabilised when both the APC/C and proteasome were inhibited (*Figure 5B and C*, green and orange lines). Complete stabilisation was not achieved through APC/C inhibition alone (*Figure 5C*, blue line). However, 80% of CCNB1 was stabilised by the combined apcin/pro-TAME treatment during the first 20 min window after CDK1 inhibition in which CCNB1 is normally fully degraded. This trend is also evident in the stability of securin and TPX2. Furthermore, Aurora A, whose destruction is CDH1-dependent, was largely stabilised by the addition of APC/C inhibitors (*Figure 5D*, blue line). The changes in CDC20 level seen here are consistent with mass spectrometry data whereby ~ 50% is degraded over 60 min (*Figure 3—figure supplement 1E*; *Figure 5B and E*, green line). Interestingly, whilst the CDC20 signal was stabilised

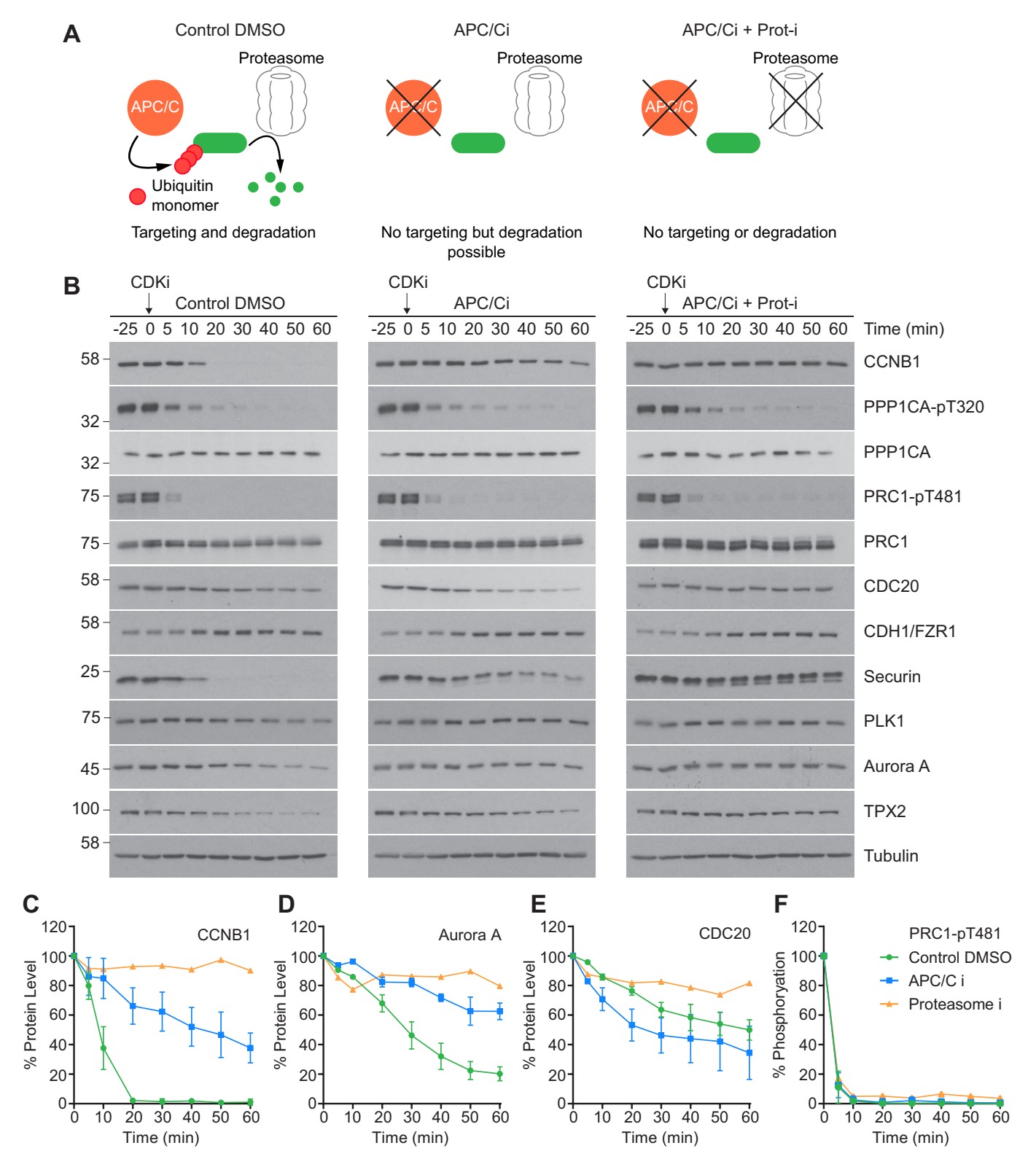

**Figure 5.** CDK inhibition in the absence of APC/C or proteasome activity triggers mitotic exit. HeLa cells were arrested in mitosis as previously described. (**A**) Schematic detailing the APC/C and proteolytic activities in each condition. (**B**) Western blot showing the protein levels and phosphorylation state in mitotically arrested HeLa cells, following CDK inhibition, in the presence or absence of APC/C inhibitors (APC/Ci), 400 μM

*Figure 5 continued on next page*

*Figure 5 continued*

apcin and 25 μM pro-TAME, or APC/C and proteasome inhibitors (Proteasome-i), 40 μM MG-132. Densitometric quantification of the levels of (**C**) CCNB1, (**D**) Aurora A, (**E**) CDC20 and (**F**) PRC1-pT481 (Mean ± SEM, Control and APC/Ci n = 2, proteasome-i n = 1).

by proteasome inhibition (*Figure 5E*, orange line), its levels were not stabilised by APC/C inhibitors alone (*Figure 5E*, blue line). This suggests that the CDC20 ubiquitination required for proteolytic destruction occurred within this experiment prior to drug addition, and is consistent with this occurring while the spindle checkpoint is active, in an APC15-dependent manner (*Mansfeld et al., 2011*; *Uzunova et al., 2012*). This is, however, inconsistent with CDC20 destruction being dependent on APC/C^CDH1 activity. Predictably, the levels of CDH1 were stable throughout the time course, as CDH1 is necessary for APC/C activity into the next G1 phase.

Importantly, under all conditions dephosphorylation of PP1 and PP2A-B55 substrates could be seen to occur as rapidly as in control cells, as indicated by the dephosphorylation kinetics of PPP1CA-pT320 and PRC1-pT481 (*Figure 5B and E*). These results support the hypothesis that phosphatase activity is responsible for the bulk of anaphase regulation. Moreover, they suggest that the only proteolytic event necessary for mitotic exit is the destruction of CCNB1, to inactivate CDK1 and relieve separase inhibition. However, a more in-depth analysis was necessary to see if this holds true on a global level. This was achieved by combining the established mass spectrometry protocol and high-resolution time course with chemical inhibition of the APC/C. APC/C inhibition resulted in stabilisation of CCNB1 and securin, especially during the first 20 min following CDK inhibition (*Figure 6D* and *Figure 6—figure supplement 1*). Interestingly, while securin migrated as a single band under control conditions a smear above a single band was observed when APC/C was inhibited (*Figure 5B* and *Figure 6—figure supplement 1*). This could indicate multiple modification states of securin, necessary for ubiquitination, which are not normally observable under control conditions (*Hellmuth et al., 2014*). Levels of CDC20 were again seen to fall slowly irrespective of APC/C inhibition (*Figure 6—figure supplement 1A and B*), supporting the view that this is not rapidly targeted by the APC/C early in mitotic exit. As before, dephosphorylation of PPP1CA-pT320 and PRC1-pT481 was equally rapid in control or APC/C-inhibited cells (*Figure 6E and F* and *Figure 6—figure supplement 1*). These samples were then subjected to mass spectrometry analysis, resulting in the clustering of 2950 phospho-sites.

It was found that five clusters gave rise to the most sensible groupings of phospho-site behaviour across both time courses (*Figure 6—source data 1*). This allowed a category to be included for noisier data which was not quantified equally well in both conditions (302 phospho-sites, not shown). The four remaining clusters followed the same pattern observed in the previous analysis (*Figure 6A and B*, *Figure 2*), with phospho-site groupings of early, intermediate, late and stable. Furthermore, the average half-life of phospho-sites within each cluster was very similar in the control and APC/C inhibited conditions (*Figure 6B*). Predictably, the sequence logo of each of these clusters very closely matched those seen in analysis of mitotic exit under conditions where the APC/C is active (*Figure 2B*). This indicates that inhibition of APC/C-dependent proteolysis neither prevents dephosphorylation during mitotic exit nor affects the order in which they occur. Furthermore, it appears that the minimal proteolytic requirement for a successful mitotic exit is the destruction of CCNB1. Together, these results are consistent with dephosphorylation, rather than proteolysis, determining the order of events during anaphase (*Afonso et al., 2019*).

## PP1 and PP2A-B55 share a preference for basic phospho-threonine sites

Previous work has demonstrated that the preference of PP2A-B55 towards basic substrates impacts the order of events in anaphase (*Cundell et al., 2016*). In the global analysis presented here the most basic phospho-sites were dephosphorylated the most rapidly, suggesting that a preference for basicity is an inherent characteristic of PPPs and a general principle of phosphatase-mediated anaphase regulation. To test whether PP1 shares this same preference, samples from a high-resolution analysis of mitotic exit under PP1 depleted conditions (*Bancroft et al., 2020*) were subjected to mass spectrometry analysis. To identify PP1-dependent clusters, the average phospho-site half-life was calculated for both control and PP1 depletion within each cluster. Clusters where phospho-site

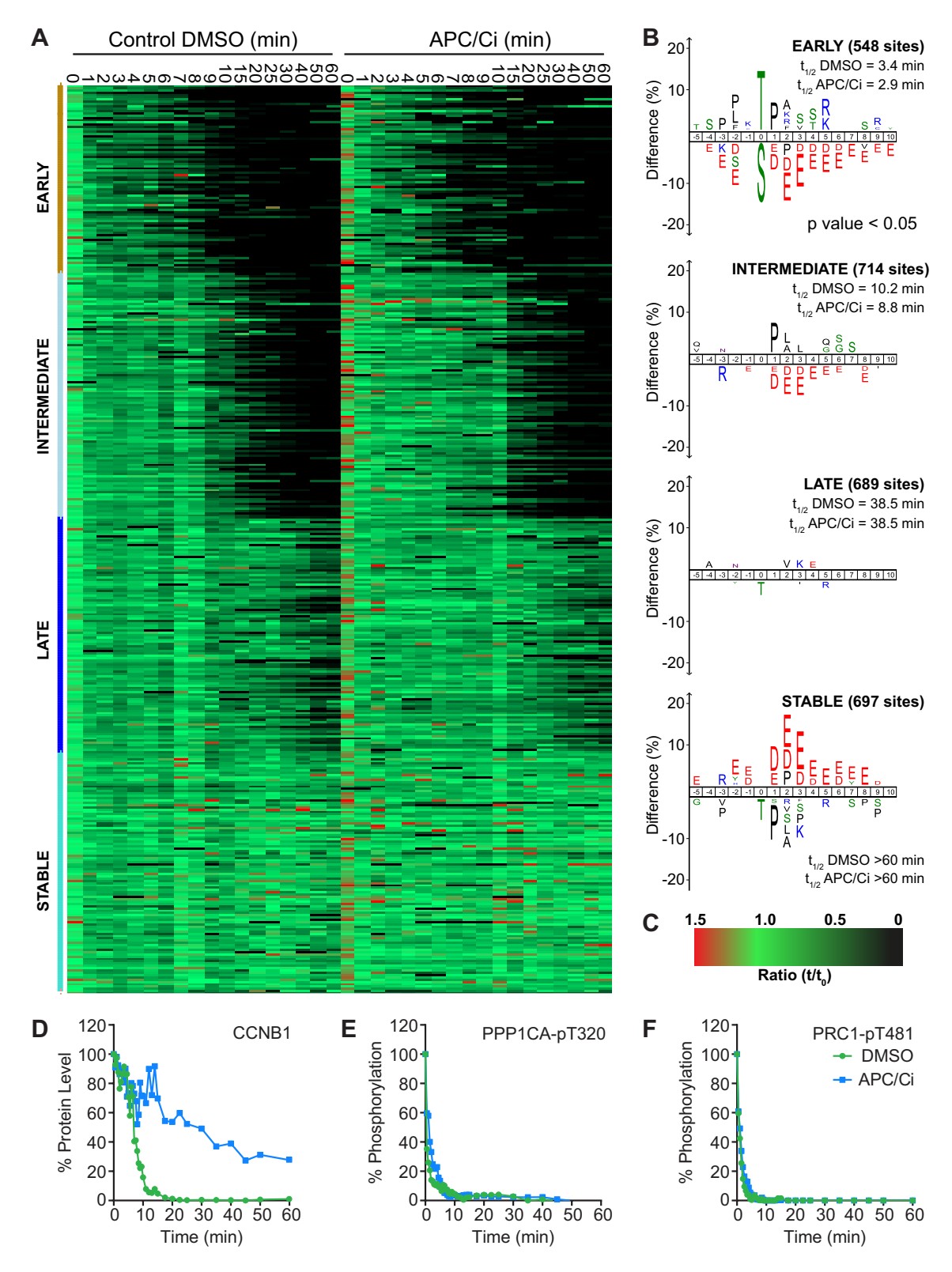

**Figure 6.** APC/C-dependent proteolysis is not necessary for ordered dephosphorylation during mitotic exit. HeLa cells were arrested in mitosis and mitotic exit triggered as previously. (**A**) Heatmap of phospho-sites clustered based on their behaviour over time, following CDK inhibition, in the presence of either DMSO or APC/C inhibitors (APC/Ci), 400 µM apcin and 25 µM pro-TAME (*Figure 6—source data 1*). Clusters are colour-coded with a bar to the left side, and named accordingly to correspond to the early, intermediate, late and stable groups described in *Figure 2*. (**B**) Sequence logo

*Figure 6 continued on next page*

*Figure 6 continued*

analysis showing the relative enrichment of residues in the adjacent cluster when compared against the sequences of all clustered phospho-peptides. Number of phospho-sites and the average half-life of each cluster is indicated. (**C**) Key linking heatmap colour with observed ratio. (**D-F**) Densitometric quantification of western blot samples (*Figure 6—figure supplement 1*), corresponding to those used for mass spectrometric analysis, is shown for (**D**) CCNB1, (**E**) PPP1CA-pT320 and (**F**) PRC1-pT481 (n = 1).

The online version of this article includes the following source data and figure supplement(s) for figure 6:

**Source data 1.** Phospho-proteome data and clustering analysis of mitotic exit triggered using CDK inhibtion under either control or APC/C inhibited conditions.

**Figure supplement 1.** High-resolution analysis of mitotic exit following APC/C inhibition.

half-life in the PP1 depleted condition was >50% longer than that of control were taken forward for further analysis. Depletion of PPP1CB had little effect on bulk dephosphorylation over time and no clusters were found to have a half-life >50% different from control (*Figure 7—source data 1*). This agrees with observations that PPP1CB is part of a specific myosin phosphatase and its depletion does not result in defects in furrowing and cytokinesis suggestive of a predominantly non-mitotic role (*Capalbo et al., 2019*; *Takaki et al., 2017*; *Zeng et al., 2010*).

The clustering comparison of 2734 phospho-sites from control and PPP1CA/C depleted samples was empirically found to require eight clusters corresponding to early, intermediate, late and stable phosphorylation sites defined in *Figure 2* (*Figure 7A*). In this analysis, the early and intermediate substrates both split into two groups which, in addition to the late substrates, all demonstrated extended half-lives when PPP1CA/C was depleted (*Figure 7B*). The sequence characteristics of these clusters followed the trends observed previously, with exclusion of acidic amino acids immediately downstream of the phosphorylation site correlating with decreased half-life of phosphorylation (*Figure 2B*). The two early clusters, half-lives <5 min in the control, reveal that PP1 has a modest preference for phospho-threonine upstream of a basic patch, and some specificity towards CDK-sites (*Figure 7B*). Many of the cohort of PP1-dependent sites we have identified are not currently assigned as substrates of a particular phosphatase (*Figure 7—source data 1*), and will be useful to explain the functions of PPP1CA/C in mitosis and mitotic exit. In agreement with the idea that PP1 lies upstream of PP2A-B55 and thus controls its activation (*Heim et al., 2015*; *Ma et al., 2016*; *Rogers et al., 2016*), some of these sites, including those on PRC1 and ENSA, are known PP2A-B55 substrates which are indirectly affected by removal of PP1 (*Cundell et al., 2016*). For the intermediate sites with half-lives <15 min in the control, the motifs show minor exclusion of phospho-threonine with similar, but slightly reduced, surrounding sequence preferences to that seen in the early clusters (*Figure 7B*). The late cluster demonstrates an exclusion of phospho-threonine with minimal sequence selectivity of the surrounding environment.

Depletion of PP1 has been reported to lead to hyperphosphorylation of Histone H3-T3 in prometaphase and metaphase (*Qian et al., 2011*; *Qian et al., 2015*). Increased steady-state phosphorylation could account for some of the reduction in the rate of anaphase dephosphorylation observed following PP1 depletion (*Figure 7*). To address this issue, the relative levels of steady-state phosphorylation were compared in control and PPP1CA/C depleted conditions (*Figure 7—figure supplements 1* and *Figure 7—source data 1*). The level of steady state phosphorylation was defined as the starting ratio at 0 min. Ratios from PP1 depleted samples were adjusted with respect to control samples according to relative phosphorylation-site intensities (Materials and methods; Analysis of mass spectrometry time course data). The majority of phosphorylation sites showed little or modest increase in abundance following depletion of PP1. These likely represent two groups, the first group are phosphorylation sites whose regulation is completely independent of PP1 activity. The second group would consist of phosphorylation sites which are dephosphorylated in a PP1-dependent manner during anaphase. Importantly, the pool of PP1 responsible for the anaphase dephosphorylation of these sites would be inhibited during mitosis. Therefore, depletion of PP1 would not result in a large change in the abundance of phosphorylation at these sites during mitosis, rather it would delay the kinetics of dephosphorylation during anaphase.

This approach identified a sub-set of ~300 phosphorylation sites where the steady state level of phosphorylation more than doubled following depletion of PP1 (*Figure 7—figure supplement 1A*, >2 fold red line). This behaviour is indicative of PP1-dependent regulation during prometaphase and metaphase, marking these phosphorylation sites as candidate substrates of PP1 activity prior to

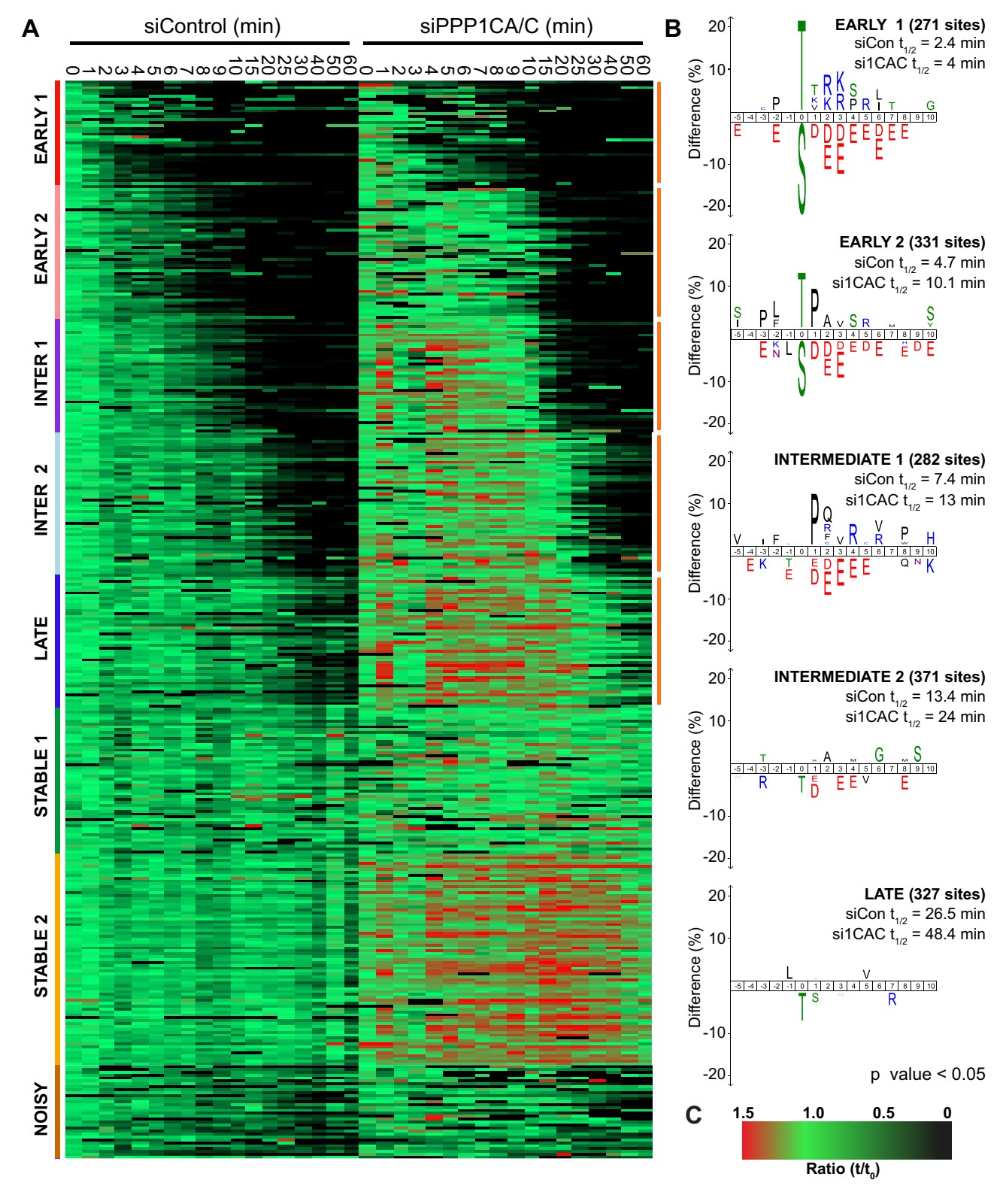

**Figure 7.** Depletion of PP1 increases the half-life of protein phosphorylation during mitotic exit. HeLa cell samples depleted of either control or PPP1CA/C were processed for mass spectrometry and analysed as described previously (*Bancroft et al., 2020*). (**A**) Clustering heat-map comparing the trends in phosphorylation over time in control and PPP1CA/C depleted cells following CDK1 inhibition (*Figure 7—source data 1*). Times shown are in minutes. Clusters are colour-coded with a bar to the left side, and named accordingly to correspond to the early, intermediate (inter), late and stable

*Figure 7 continued on next page*

*Figure 7 continued*

groups described in *Figure 2*. Clusters where the average siPPP1CA/C half-life is >50% more than that of siControl are marked with an orange line to the right of the heatmap. (B) Sequence logo analysis showing the relative enrichment of residues in the correspondingly labelled cluster when compared against the sequences of all clustered phospho-peptides. Number of phospho-sites and the average half-life of each cluster is indicated. (C) Key linking heatmap colour with observed MS ratio.

The online version of this article includes the following source data and figure supplement(s) for figure 7:

**Source data 1.** Phospho-proteome data and clustering analysis of mitotic exit triggered using CDK inhibtion under either control or PP1 depleted conditions.

**Figure supplement 1.** Depletion of PP1 increases the steady-state level of a subset of mitotic phosphorylations.

anaphase (*Figure 7—source data 1*). These sites include BUB1-pS459, which forms part of a pseudo-MELT motif to promote mitotic checkpoint complex assembly (*Ji et al., 2017*), in addition to multiple phosphorylations on INCENP and Ki-67 (*Figure 7—figure supplement 1A–D*). Not all phosphorylation sites detected on the same protein demonstrated the same behaviour (*Figure 7—figure supplement 1A–D*). These data suggest that the requirement for PP1-dependent regulation during prometaphase is limited to a subset of phosphorylation sites, and may relate to the function of each individual site. Interestingly, Ki-67 is itself a binding partner of PP1 (*Booth et al., 2014*), however the impact of phospho-regulation at these sites on the localisation and function of Ki-67 during mitosis and the nature of the PP1 holoenzyme which mediates this is unknown. Importantly, these PP1-dependent changes occurred on both phospho-threonine and phospho-serine.

We conclude that the preference for phospho-threonine in the absence of downstream negative charge may be a general property of PPP holoenzymes with relevance for the ordering of events during mitotic exit. Together, these results support the view that the cascade of dephosphorylation reactions mediated by a combination of PP1 and PP2A provides a mechanism to determine the order of events during anaphase.

## Discussion

### Proteolytic and phosphatase control of mitotic exit

The combination of high temporal-resolution mass spectrometry with the cell biological techniques used in this research provides insights into cell cycle control. These data show that proteolysis plays a crucial role initiating the mitotic exit transitions whereas phosphatase activity is required for downstream regulation of order within the system. Therefore, the following model is proposed with regard to how CDK-inhibited phosphatases and the APC/C co-operate to ensure orderly progression through mitotic exit (*Figure 8*). Proteolysis of a single substrate, cyclin B, triggers a graded series of dephosphorylations whose rate is encoded into the system at the level of individual phospho-sites (*Bouchoux and Uhlmann, 2011*). Due to differential regulation by CDK1-cyclin B1, in mammalian cells multiple thresholds of cyclin B destruction ensure ordered activation of CDK-inhibited phosphatases which imparts further timing properties to the system. Once both PP1 and PP2A-B55 are active, phosphorylations at thousands of sites can be removed helping to reset the system for the next cell cycle. The order of these dephosphorylations, and therefore anaphase events, is determined by the suitability of each substrate for interactions with a phosphatase active site, dependent upon the chemical and physical properties of the amino acid constituents of each phospho-site (*Figure 8*). Finally, late in telophase the APC/C targets many mitotic proteins for destruction, preparing the cell for the subsequent G1 phase.

### Ordered proteolysis of cyclins during mitosis and mitotic exit

Here, we have primarily focused on CCNB1 in the context of driving mitotic exit, however both CCNA2 and CCNB2 play important roles during mitosis. CCNA2 is sufficient to trigger mitotic entry in the absence of CCNB1, and can form active CDK1-CCNA2 complexes (*Desai et al., 1995*; *Furuno et al., 1999*; *Gong and Ferrell, 2010*; *Hégarat et al., 2020*; *Swenson et al., 1986*). Prometaphase destruction of CCNA2 is required for successful segregation of chromosomes and timely mitotic exit (*Kabeche and Compton, 2013*). We would expand upon this by suggesting that prior destruction of CCNA2 ensures that proteolysis of CCNB1 creates a state of low CDK activity,

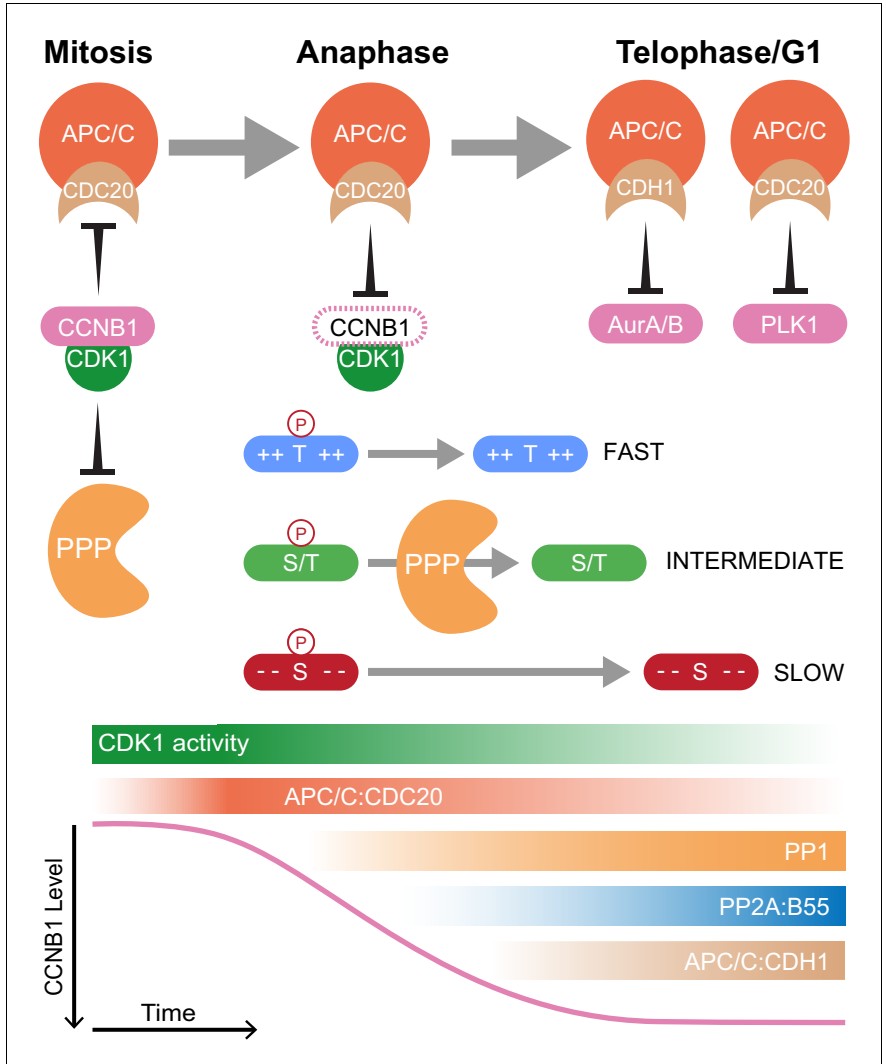

**Figure 8.** Selective proteolysis of Cyclin B1 initiates ordered dephosphorylation to drive mitotic exit. Model highlighting the distinct roles of ubiquitin-mediated proteolysis and phospho-protein phosphatases (PPP) in the regulation of mitotic exit. The activities depicted in the lower panel scale from maximum (dark shading) to non-zero, minimum (light shading) values. During metaphase, high levels of CDK1:CCNB1 activity maintains the inhibition of PP1 and PP2A-B55 as well as APC/C^CDC20. This ensures that mitotic phosphorylations are protected and cells cannot leave mitosis prematurely, before the proper alignment and attachment of chromosomes. Following spindle checkpoint silencing, APC/C^CDC20 can target CCNB1 for destruction leading to gradual CDK1 inactivation and sister chromatid separation. As CDK1 activity falls, inhibition of PP1 and PP2A-B55 is relieved triggering a cascade of dephosphorylations. The rate of dephosphorylation is encoded into the system based on the charge characteristics of each phosphorylation site and their suitability for interaction with the PPP active site, as determined by amino acid sequence. These dephosphorylations co-ordinate the many, rapid changes which must occur to create two new daughter cells. Later, in telophase or G1, APC/C^CDC20 and APC/C^CDH1 target the remaining the mitotic factors for destruction to reset the cell for the subsequent G1 phase. The late-APC/C substrates include accessory mitotic kinases, such as Aurora A/B and PLK1, and kinetochore proteins, such as BUBR1 and CENPF.

fundamentally different to G2 which is governed by CDK1-CCNA2 activity. This inherent asymmetry would drive the cell cycle forward into anaphase rather than reverting back to a G2-like state (*Holder et al., 2019*). Consistent with this, expression of a non-degradable form of CCNA2 traps cells in anaphase A and does not permit further progression through mitotic exit (*Geley et al., 2001*).

The role of CCNB2 is less clear, with CCNB2 deficient mice developing normally into fertile adults (*Brandeis et al., 1998*). Some studies in cells have proposed that CCNB2 can compensate, at least partially, for CCNB1 (*Bellanger et al., 2007*; *Gong and Ferrell, 2010*; *Hégarat et al., 2020*). One possible explanation for these observations is that, in the absence of CCNB1, CCNB2 in combination with the declining levels of CCNA2 can provide the necessary CDK1 activity to complete mitosis. Furthermore, while levels of CCNB2 do decrease during our analysis of mitotic exit, CCNB2 is present at lower concentrations in cells (*Brandeis et al., 1998*). This is supported by the inconsistent detection of CCNB2 between time points (*Figure 2—source data 1* and *Figure 3—source data 1*). We therefore propose that it is the destruction of solely CCNB1 which drives mitotic exit.

## APC/C fulfils two temporally distinct roles during mitotic exit

The very different rates of proteolysis between early and late APC/C substrates described here supports the view that CDC20 and CDH1 globally satisfy two different requirements of mitotic exit (*Figures 3* and *8*). Firstly, creation of a cell state free from CDK-cyclin activity, through destruction of CCNB1, to permit mitotic exit and phosphatase activation. Although, securin and GMNN were also observed as early substrates of the APC/C, we would argue that destruction of CCNB1 is the primary function of the APC/C at this time. Cells can survive without securin, as CCNB1 can inhibit separase both directly and through promoting CDK1 activity (*Gorr et al., 2005*; *Huang et al., 2005*; *Stemmann et al., 2006*). Moreover, loss of CDK activity is required for cells to capitalise on the DNA replication licensing window created during anaphase, between the sequential destruction of GMNN and CDT1 (*Ballabeni et al., 2004*; *Clijsters et al., 2013*). The second function of the APC/C is to promote destruction of mitotic factors and reset the cellular state for the subsequent cell cycle. Importantly, the proposed two-stage model of APC/C activity ensures temporal separation between the global inactivation of CDK1-cyclin B and proteolysis of Aurora A, Aurora B and PLK1, a requirement for creation of the anaphase state (*Holder et al., 2019*). Activity of Aurora A, Aurora B, PLK1 and pools of CDK1-Cyclin B1 therefore persist later into mitotic exit (*Figure 2—source data 1*, *Figure 3—source data 1*, *Figure 4D* and *Figure 5*; *Afonso et al., 2019*; *Floyd et al., 2008*; *Lindon and Pines, 2004*; *Mathieu et al., 2013*). This localised kinase activity then regulates the successful completion of chromosome separation, cytokinesis and abscission (*Afonso et al., 2019*; *Afonso et al., 2014*; *Bastos and Barr, 2010*; *Bastos et al., 2014*; *Nunes Bastos et al., 2013*; *Steigemann et al., 2009*).

A simple view would be to separate these functions based on APC/C coactivator, CDC20 for cyclin destruction and CDH1 for resetting G1 cell status. This is supported by a proteomic analysis of meiotic exit in *Xenopus* eggs which found that wide-spread dephosphorylation was triggered by the destruction of a small number of proteins, typified by the APC/C[CDC20] substrates: CCNB1/2, securin and GMNN (*Presler et al., 2017*). These eggs lack CDH1, therefore typical APC/C[CDH1] substrates such as Aurora kinases were, unsurprisingly, stable. The absence of CDH1-dependent destruction is thought to help bypass G1 and promote rapid divisions within the early embryo (*Presler et al., 2017*), highlighting a role for APC/C[CDH1] activity in promoting the formation of a stable G1 state. Nevertheless, APC/C[CDC20] is required for the destruction of BUBR1, CENPF and PLK1 late in anaphase (*Choi et al., 2009*; *Floyd et al., 2008*; *Gurden et al., 2010*). Moreover, we show that CDC20 is not rapidly degraded during mitotic exit (*Figure 5*, *Figure 3—figure supplement 1* and *Figure 6—figure supplement 1*) suggesting that both CDC20 and CDH1 function during late anaphase and telophase to degrade mitotic proteins (*Figure 8*). At present it is unclear how the APC/C promotes such distinct proteolytic kinetics between early and late substrates.

## Proteolytic regulation of CDC20 during mitotic exit

CDC20 is unique in that it functions as both an activator and inhibitor of the APC/C and is therefore subject to complex regulation through post-translational modification. Previous work suggested that continuous synthesis and destruction of CDC20 was necessary to maintain the mitotic state and allow for rapid APC/C activation following SAC silencing (*Foe et al., 2011*; *Foster and Morgan, 2012*; *Nilsson et al., 2008*; *Varetti et al., 2011*; *Wang et al., 2017*). The data presented here argue against a requirement for protein synthesis during mitosis in mammalian cells (*Figure 4* and *Figure 4—figure supplement 1*). Inhibition of protein synthesis did not cause cells to leave mitosis prematurely, nor did this inhibition override checkpoint-mediated mitotic arrest. Evidence suggests that

in *S. cerevisiae*, APC/C$^{CDC20}$ activity inversely correlates with levels of CDC20 transcription (*Wang et al., 2017*). Nevertheless, in mammalian cells there is no known feedback between APC/C activity and protein expression, which would be necessary to ensure the correct levels of CDC20 and cyclin B synthesis are maintained in spite of continued destruction. Likewise, mammalian cells inhibit bulk protein synthesis at mitotic entry whilst budding yeast manufacture proteins throughout their cell cycle (*Polymenis and Aramayo, 2015*). These variations in regulation of protein expression, alongside the fundamental differences between open and closed mitosis could explain the divergent requirements for continued CDC20 synthesis during mitosis in budding yeast and mammalian cells. The length and nature of mitotic arrest used to assess the requirement of CDC20 synthesis could account for discrepancies in results between different studies. Synthesis of mitotic factors such as CDC20 and CCNB1 is likely to be more important for maintenance of spindle checkpoint activity over periods of prolonged mitotic arrest (*Mena et al., 2010*).

CDC20 destruction is thought to be a consequence of APC/C switching coactivators to CDH1, following CDK1 inactivation. Here we show that CDC20 destruction was independent of ongoing APC/C activity during mitotic exit, suggesting that the necessary ubiquitination of CDC20 occurs prior to checkpoint silencing (*Figure 5* and *Figure 6—figure supplement 1*). These findings are consistent with CDC20 being ubiquitinated as part of APC/C$^{CDC20}$-bound MCC complex (*Foe et al., 2011*; *Mansfeld et al., 2011*; *Uzunova et al., 2012*). Given that ~ 50% of CDC20 still remained one hour after CDK inhibition, it is important to understand whether any particular pools of CDC20 are preferentially degraded during mitotic exit.

## CDK1-inhibited phosphatases are activated with distinct timings to co-ordinate the events of early anaphase

PP1 and PP2A-B55 carry out distinct roles to ensure proper passage into mitotic exit; APC/C activation and initiation of cytokinesis, respectively (*Cundell et al., 2013*; *Cundell et al., 2016*; *Wu et al., 2009*). These functions explain the requirement for two CDK1-inhibited phosphatases that are divergently regulated. Direct CDK1 inhibition, in combination with phosphorylation of regulatory interactors, ensures that the bulk of PP1 is inactive, allowing a global increase in phosphorylation status upon mitotic entry (*Wu et al., 2009*). As cyclin B is degraded, PP1 rapidly responds globally to the changes in CDK1 activity becoming dephosphorylated (*Figures 1* and *2* and *Figure 2—figure supplement 1*; *Wu et al., 2009*). PP2A-B55 inhibition is maintained during this period by the BEG pathway. Once both PP1 and the required interactors are dephosphorylated, PP1 can then oppose the CDK1 phosphorylation of MASTL and initiate PP2A-B55 activation (*Rogers et al., 2016*). Thus, two waves of phosphatase activity co-ordinate the early events of mitotic exit. This is reflected in the mass spectrometry time course data following MPS1 inhibition. PPP1CA-pT320 dephosphorylation commenced ~5 min earlier than the model PP2A-B55 substrate PRC1-pT481. Moreover, the half-life of PRC1-pT481 was ~2 min longer than that of PPP1CA-pT320 (*Figure 2—figure supplement 1G*). These data are consistent with the delay of ~2–4 min observed between chromosome separation and commencing central spindle formation (*Cundell et al., 2013*). Once activated, both PP1 and PP2A-B55 lead to the dephosphorylation of many substrates during mitotic exit (*Figure 8*; *Cundell et al., 2016*; *Wu et al., 2009*). Dephosphorylation rate correlates with the sequence characteristics of each substrate and this trend applies to the specificity of the individual phosphatases PP1 (*Figure 7A*) and PP2A-B55 (*Cundell et al., 2016*), as well as to the cell as a whole (*Figure 2* and *Figure 2—figure supplement 1*). These preferences may relate to the conserved acidic groove found in PP1 and PP2A catalytic subunits close to the active site, providing a molecular basis for the discrimination against more acidic phospho-sites (*Figure 8*). The negative charge provided by acidic residues in addition to a phosphate group would be electrostatically repelled from this binding groove, preventing efficient substrate binding and dephosphorylation (*Figure 8*).

Regulated phosphatase activities and highly selective proteolysis thus initiate and choreograph the rapid series of events which occur during mitotic exit and ensure that faithful inheritance of genetic material is coupled to the formation of new cells. Our data provide evidence in support of this conclusion and will support future studies in this area.

# Materials and methods

**Key resources table**

| Reagent type (species) or resource | Designation | Source or reference | Identifiers | Additional information |
|---|---|---|---|---|
| Cell line (*Homo sapiens*) | HeLa S3 | ATCC | ID_source:CCL-2.2; RRID:CVCL_0058 | Mycoplasma negative |
| Cell line (*Homo sapiens*) | HeLa S3: CCNB1-GFP | *Alfonso-Pérez et al., 2019*; *Hayward et al., 2019* | | Mycoplasma negative Generated from RRID:CVCL_0058 HeLa S3 cells. Validated through sequencing around CCNB1 locus, Western blot and imaging, both fixed and live cell. |
| Antibody | Anti-Aurora A (rabbit monoclonal) | Cell Signalling | ID_source:4718S; RRID:AB_2061482 | WB (1:2000) |
| Antibody | Anti-Aurora B (mouse monoclonal) | BD Transduction Labs | ID_source:611083; RRID:AB_398396 | WB (1:1000) |
| Antibody | Anti-BUBR1 (rabbit polyclonal) | Bethyl | ID_source:A300-386A; RRID:AB_386097 | WB (1:2000) |
| Antibody | Anti-CDC20 (rabbit polyclonal) | Proteintech | ID_source:10252–1; RRID:AB_2229016 | WB (1:1000) |
| Antibody | Anti-CDH1/fzr (mouse monoclonal) | Santa Cruz | ID_source:sc-56312; RRID:AB_783404 | WB (1:500) |
| Antibody | Anti-Cyclin B1 (mouse monoclonal) | Millipore | ID_source:05–373; RRID:AB_309701 | WB (1:5000) |
| Antibody | Anti-MAD2 (sheep polyclonal) | Barr/Gruneberg Labs. This paper. | Raised in sheep (Scottish Blood Transfusion Service) using full length human 6His-MAD2 expressed in bacteria and coupled to KLH as the antigen. The serum was affinity purified using GST-tagged MAD2. | WB (1:1000) |
| Antibody | Anti-PLK1 (goat polyclonal) | *Neef et al., 2003* | | WB (1:2000) |
| Antibody | Anti-PPP1CA (rabbit polyclonal) | Bethyl | ID_source:A300-904A; RRID:AB_2284190 | WB (1:1000) |
| Antibody | Anti-PPP1CA-pT320 (rabbit monoclonal) | AbCam | ID_source:ab62334; RRID:AB_956236 | WB (1:2000) |
| Antibody | Anti-PRC1 (rabbit polyclonal) | *Gruneberg et al., 2006* | | WB (1:2000) |
| Antibody | Anti-PRC1-pT481 (rabbit monoclonal) | AbCam | ID_source:EP1514Y; ab62366; RRID:AB_944969 | WB (1:4000) |
| Antibody | Anti-PRC1-pT602 (rabbit polyclonal) | *Neef et al., 2007* | | WB (1:1000) |
| Antibody | Anti-Securin (rabbit monoclonal) | AbCam | ID_source:EPR3240; ab79546; RRID:AB_2173411 | WB (1:1000) |
| Antibody | Anti-TPX2 (mouse monoclonal) | AbCam | ID_source:ab32795; RRID:AB_778561 | WB (1:2000) |
| Antibody | Anti-tubulin (mouse monoclonal) | Sigma | ID_source:T6199; Clone DM1A; RRID:AB_477583 | WB (1:10,000) |
| Antibody | Anti-mouse 2°-HRP (donkey polyclonal) | Jackson | ID_source:715-035-150; RRID:AB_2340770 | WB (1:2000) |
| Antibody | Anti-rabbit-2°-HRP (donkey polyclonal) | Jackson | ID_source:711-035-152; RRID:AB_10015282 | WB (1:2000) |
| Antibody | Anti-Sheep 2°-HRP (donkey polyclonal) | Jackson | ID_source:713-035-147; RRID:AB_2340710 | WB (1:2000) |
| Aeptide, recombinant protein | Lysyl-endopeptidase | Wako Pure Chemical Industries | ID_source:121–05063 | |

*Continued on next page*

*Continued*

| Reagent type (species) or resource | Designation | Source or reference | Identifiers | Additional information |
|---|---|---|---|---|
| Peptide, recombinant protein | Trypsin Gold | Promega | ID_source:V5280 | |
| Chemical compound, drug | Apcin | Tocris | ID_source:5747/10 | Resuspend in DMSO to 50 mM for stock |
| Chemical compound, drug | AZ3146 | Tocris | ID_source:3994/10 | Resuspend in DMSO to 5 mM for stock |
| Chemical compound, drug | Cycloheximide | Cell Signalling | ID_source:2112S | Resuspend in DMSO to 100 mM for stock |
| Chemical compound, drug | Flavopiridol | Tocris | ID_source:3094/10 | Resuspend in DMSO to 5 mM for stock |
| Chemical compound, drug | MG-132 | CalBiochem | ID_source:47490 | Resuspend in DMSO to 20 mM for stock |
| Chemical compound, drug | Nocodazole | CalBiochem | ID_source:487928 | Resuspend in DMSO to 200 µg/ml for stock |
| Chemical compound, drug | Okadaic Acid | Enzo | ID_source:ALX-350–003 | Resuspend in DMSO to 500 µM for stock |
| Chemical compound, drug | Pro-TAME | Boston Biochem | ID_source:I-440 | Resuspend in DMSO to 20 mM for stock |
| Chemical compound, drug | Taxol | CalBiochem | ID_source:580555 | Resuspend in DMSO to 200 µg/µl for stock |
| Chemical compound, drug | Thymidine | CalBiochem | ID_source:6060 | Resuspend in Millipore filtered water to 100 mM for stock. |
| Sequence-based reagent | siControl | *Bancroft et al., 2020*, Dharmacon | | Custom Sequence 5' cguacgcggaauacuucgauu |
| Sequence-based reagent | siPPP1CA | *Bancroft et al., 2020*, Dharmacon | ID_source: NM_002708.3 n | Custom Sequence 5' uggauugauuguacagaaauu |
| Sequence-based reagent | siPPP1CB | *Bancroft et al., 2020*, Dharmacon | ID_source: NM_002709.2 | Custom Sequence 5' gggaagagcuuuacagacauu |
| Sequence-based reagent | siPPP1CC | *Bancroft et al., 2020*, Dharmacon | ID_source: NM_002710.3 | Custom Sequence 5' gcggugaaguugaggcuuauu |
| Commercial assay or kit | Protein Assay Dye-Reagent Concentrate | Bio-Rad | ID_source:5000006 | |
| Commercial assay or kit | ECL Blotting Reagents | GE Healthcare | ID_source:GERPN2109 | |
| Software, algorithm | Adobe Illustrator CS4 | Adobe Systems Inc | ID_source:RRID:SCR_010279 | |
| Software, algorithm | Adobe Photoshop CS4 | Adobe Systems Inc | ID_source:RRID:SCR_014199 | |
| Software, algorithm | Fiji 1.52 p | National Instiues of Health, USA | ID_source:RRID:SCR_002285 | |
| Software, algorithm | IceLogo 1.2 | *Colaert et al., 2009* | ID_source:RRID:SCR_012137 | |
| Software, algorithm | MaxQuant | *Tyanova et al., 2016a* | ID_source:RRID:SCR_014485 | |
| Software, algorithm | Perseus | *Tyanova et al., 2016b* | ID_source:RRID:SCR_015753 | |
| Software, algorithm | Prism 8.3.1 | GraphPad Software | ID_source:RRID:SCR_002798 | |
| Other | C18 Discs | Empore | ID_source:3M2215 | |
| Other | Phospho-peptide Enrichment TopTip | Glygen | ID_source:TT1TIO | |
| Other | SepPak Reverse-Phase C18 Columns | Waters | ID_source:WAT023501 | |

## Cell culture

HeLa cells were grown in DMEM containing 10% [vol/vol] FBS at 37°C and 5% $CO_2$. CRISPR/Cas9-edited HeLa cells, with an inserted GFP tag in the C-terminus of the CCNB1 gene product, were grown in the same medium (*Alfonso-Pérez et al., 2019*; *Hayward et al., 2019*).

## High-resolution mitotic exit time courses for western blot and mass spectrometry

For high-resolution time courses HeLa cells were arrested in mitosis using 100 ng/ml nocodazole treatment for 20 hr. Mitotic cells were harvested by shake-off, washed twice in 1x PBS, and once in Opti-MEM, both of which had been pre-equilibrated to 37°C, 5% $CO_2$. Cells were counted and resuspended in pre-equilibrated Opti-MEM to give 15,000,000 cells/ml and incubated for 25 min (37°C, 5% $CO_2$) to allow cells to rebuild bipolar mitotic spindles. CDK inhibition, using 20 µM flavopiridol, was used to trigger mitotic exit. Where necessary, drugs were added at the start of the 25 min incubation to ensure maximal inhibition prior to the addition of flavopiridol. MG-132, a proteasome inhibitor, was used at 40 µM while apcin and proTAME, APC/C inhibitors, were used at 400 µM and 25 µM, respectively. Cycloheximide (CHX) was used at 100 µg/ml.

For cell lysis, 100 µl samples of cells were taken at each timepoint and added to a tube pre-aliquoted with 200 µl ice cold urea buffer (8 M urea, 100 mM ammonium-bicarbonate (AMBIC)) and immediately snap frozen. The urea buffer was supplemented with protease inhibitors (Sigma cocktail C, 1:250 dilution), phosphatase inhibitors (Sigma cocktail 3, 1:100 dilution) and 100 nM okadaic acid. For the subsequent di-methyl labelling protocol greater quantities of $t_0$ sample were required, therefore twelve 300 µl aliquots were taken at this time and each added to a tube pre-aliquoted with 600 µl ice cold urea buffer. Samples were thawed on ice for 20 min with intermittent vortexing and the concentration determined by Bradford assay. A 50 µl aliquot of lysate was taken for western blotting, added to 25 µl of 3x sample buffer and boiled for 5 min prior to loading for western blots. Typically, 6 µg was loaded on each western blot but when necessary increased to 10 µg. Western blot data from time course samples depleted of PP1 subunits were published previously (*Bancroft et al., 2020*), here the remaining lysate was subjected to trypsin digestion for subsequent proteomic analysis. Low-resolution time courses for western blot were set up as above, however cells were lysed directly in 3x sample buffer. These samples were diluted 1:2 with 1x sample buffer prior to western blotting.

## Digestion of samples for proteomic analysis

Time-course samples were reduced using 4 mM dithiothreitol (DTT) (Fluka) for 25 min at 56°C followed by alkylation, using 8 mM iodoacetamide (IAA) incubation, in the dark for 30 min. Excess IAA was quenched by addition of DTT to a final concentration of 8 mM. Proteins were digested first with lysyl endopeptidase (Lys-C) (Wako Pure Chemical Industries) for 4 hr at 37°C in 8 M urea. The sample was then diluted to 2 M urea, with 50 mM ammonium bicarbonate. Trypsin was then added for 12 hr at 37°C. Both enzymes were used at 1:50 ratio relative to the starting amount of protein. Digestions were quenched by acidification with 5% (vol/vol) formic acid (FA).

## Di-methyl labelling and titanium dioxide phospho-peptide enrichment

Digested peptides, equivalent to 0.4–1.5 mg of total protein, were bound to SepPak reverse-phase C18-columns (Waters) and subjected to on-column dimethyl labelling as previously described (*Boersema et al., 2009*). Peptides from all timepoints of the dephosphorylation assay were labelled with cyanoborohydride and deuterated formaldehyde to generate a mass increase of 32 Da per primary amine, referred to as the 'heavy' label. Additionally, 6–8 mg of 0 min timepoint was labelled with formaldehyde and cyanoborohydride to generate a mass increase of 28 Da per primary amine, referred to as the 'light' label; 1–2 mg of 0 min was labelled 'heavy'. After this, 225 µg 'light'- and 'heavy'- labelled peptides were mixed (450 µg total). A 50 µg aliquot of the peptide mix was dried under vacuum and re-suspended in 5% [vol/vol] formic acid and 5% [vol/vol] DMSO, this represented the total proteome sample. The remainder (400 µg) was subjected to titanium dioxide enrichment.

Phospho-peptide enrichment was performed using microspin columns packed with titanium dioxide (TopTip; Glygen). All spin steps were performed at 550 rpm, equivalent to 34x $g_{av}$, for 5 min at room temperature (Eppendorf Centrifuge 5427 R). Columns were washed with 65 µl elution buffer

(5% ammonia solution in water), then three times with 65 µl loading buffer (5% [vol/vol] trifluoroacetic acid (TFA), 1 M glycolic acid, 80% [vol/vol] acetonitrile (ACN)). An equal volume of loading buffer was added to the dimethyl labelled peptide mixtures. Phospho-peptides were then bound to the column 65 µl at a time. Following loading, columns were washed once each with loading buffer, 0.2% [vol/vol] TFA acid in 80% [vol/vol] ACN, followed by 20% [vol/vol] ACN. Phospho-peptides were eluted with three washes of 20 µl elution buffer (5% ammonium solution) into 20 µl of 10% [vol/vol] FA and 10% [vol/vol] DMSO. This gave a total volume of 120 µl of which 20 µl was loaded onto the mass spectrometer.

## High pH fractionation of total proteome samples

Home-made in stage tips, constructed from five 4G C18 discs (Empore) were used for fractionation, all spins were carried out at 3000x $g_{av}$, 3 min unless otherwise specified. Tips were conditioned once each with methanol, 20 mM $NH_3$/80% [vol/vol] ACN and 20 mM $NH_3$. Next 25 µg of peptides, previously resuspended in 5% FA, 5% DMSO [vol/vol], were loaded on each tip; 1500x $g_{av}$, 3 min. The flow-through was reapplied twice more and the final flow-through (FT) retained. Bound peptides were washed with 20 µl 20 mM $NH_3$ and the flow through was then pooled with FT to create the FT/ F1 fraction. Peptides were then eluted with 20 µl of 20 mM $NH_3$ buffers of increasing ACN composition (4–80%), which formed fractions F2-7. Samples containing <10% ACN were diluted 1:1 whereas samples with higher concentrations of ACN were diluted to 10% [vol/vol] ACN and 80% of the sample loaded directly on the mass spectrometer. These dilutions were performed with 5% [vol/vol] FA, 5% [vol/vol] DMSO. Fractions FT/F1 and F6-7 were run on 30 min gradients whilst fractions F2-5 were run on 120 min gradients.

## Online nano-LC and tandem mass spectrometry

LC was performed using an EASY-nano-LC 1000 system (Proxeon) in which peptides were initially trapped on a 75 µm internal diameter guard column packed with Reprosil-Gold 120 C18, 3 µm, 120 Å pores (Dr. Maisch GmbH) in solvent A (0.1% [vol/vol] FA in water) using a constant pressure of 500 bar. Peptides were then separated on a 45°C heated EASY-Spray column (50 cm × 75 µm ID, PepMap RSLC C18, 2 µm; Thermo Fisher Scientific) using a 3 hr linear 8–30% (vol/vol) acetonitrile gradient and constant 200 nl/min flow rate. Titanium dioxide-enriched peptides were introduced via an EASY-Spray nano-electrospray ion source into an Orbitrap Elite mass spectrometer (Thermo Fisher Scientific). Other samples were injected into Q-exactive mass spectrometers (Thermo Fischer Scientific), using linear 8–30% [vol/vol] acetonitrile gradients between 1 and 3 hr in length. For non-fractionated total protein samples, 6 µg was loaded onto the mass spectrometer for a 3 hr gradient. Spectra were acquired with resolution 30,000, m/z range 350–1,500, AGC target $10^6$, maximum injection time 250 ms. The 20 most abundant peaks were fragmented using CID (AGC target $5 \times 10^3$, maximum injection time 100 ms) or ETD (AGC cation and anion target $5 \times 10^3$ and $2 \times 10^5$, respectively, maximum injection time 100 ms, normalised collision energy 35%) in a data-dependent decision tree method for titanium dioxide-enriched peptides. Peaks from other samples were fragmented using CID alone. Peptide identification and quantification of 'heavy' to 'light' phospho-peptide ratios was then performed using MaxQuant (*Tyanova et al., 2016a*).

## Analysis of mass spectrometry time course data

The MaxQuant (*Tyanova et al., 2016a*) search output was analysed in Perseus (*Tyanova et al., 2016b*) as follows. Firstly, data were filtered for contaminants; identified by modification site only and identified in the reverse database only. Phospho-peptides were also filtered so that only those with a localisation score of >0.75 remained. Missing values were then replaced. If a single value was missing, it was replaced either by the average of the two adjacent values in the time series or the adjacent value if either the first or last timepoint was missing. However, two or more consecutive missing values were not replaced as these were more likely to represent a true 0 value rather than just a missed quantification. Missing value replacement was performed in Microsoft Excel. Next, the $t/t_0$ for each timepoint was averaged across repeat experiments, presuming at least two valid quantification values were observed. Protein groups and phospho-peptides were then filtered for the number of values observed over time, using CCNB1 and PRC1-pT481 as reference profiles of proteolysis and dephosphorylation, respectively. While there were a small number of dephosphorylations that

were more rapid than the PRC1-pT481, these were less well understood and would not serve as a reliable reference. At this point any remaining missing values were replaced with the constant value, 0, to enable the clustering algorithm to function correctly. The hierarchical clustering algorithm within Perseus was used, grouping rows into four clusters and outputting a heat map. For clustering a single condition, ten iterations and one restart were allowed, this was increased to 25 iterations and two restarts when comparing two conditions.

When comparing time-courses obtained under two or more different conditions it was necessary to account for changes in the steady state level of a protein or phospho-peptide using a crosswise comparison. To do this, samples comprised of equal quantities of the starting light and heavy labelled $t_0$ sample from different experimental conditions were mixed together, for example control $t_0$ light label mixed with the $t_0$ heavy label from a perturbed condition. The ratios were extracted from the cross-mixed samples and used to adjust the protein/phospho-peptide ratios relative to control. Following this, a more iterative approach to clustering was required; starting from three, the number of clusters was gradually increased. This was continued until the settings were found which yielded the most clusters without producing random groupings containing very few identifications. Taking this approach allowed the comparison of complex trends found within the large data sets from the two conditions. The heatmap shown was derived from the majority result of the clustering analysis in each case. In the case of PPP1CA/C depleted time course, an error occurred during mixing heavy and light label resulting in all ratios being up-shifted. To account for this, the average ratio from total proteome siControl and siPPP1CA/C cross-mixed samples was used to adjust the ratios from the corresponding phospho-proteome samples. These corrected ratios were then used to adjust siPPP1CA/C timecourse ratios relative to siControl as described above.

Clusters were exported separately from Perseus. The phospho-site sequences of each individual cluster were then aligned in IceLogo 1.2 software (*Colaert et al., 2009*), using the sequences of all phospho-sites clustered as the negative set. For the percentage difference in residue enrichment to be seen in the sequence logo a p-value<0.05 was required. To determine half-lives for each cluster, the average ratio and standard error of the mean, for each timepoint, was calculated. These were adjusted to a starting ratio of one, plotted and fitted with a non-linear plateau with one phase decay curve in GraphPad Prism. The half-life was the point where the curve fit crossed y = 0.5. Previously published time course proteomic data of mitotic exit under conditions of MPS1 inhibition was reanalysed here as described above (*Hayward et al., 2019*).

To calculate the half-lives of individual proteins from fractionated time course samples the Maxquant output was filtered and missing values replaced, as above. The ratios for each time point were averaged across experimental repeats and then median subtracted. A correction of 0.85 was added to all ratios to remove any negative values. To account for technical variation in detection between the samples at 50 and 60 min, where the latter value was consistently higher across all proteins, these ratios were averaged. This endpoint value was then used to calculate a percentage of protein remaining after 60 min and converted to a half-life.

## Time courses assessing contribution of protein synthesis to mitosis

HeLa cells were seeded at 50,000 cells/well on six well plates in DMEM/FBS and grown for 72 hr. These cells were arrested in mitosis using either a 100 ng/ml nocodazole, 20 hr or 85 ng/ml taxol, 16 hr. Cells for $t_0$ were collected by shake-off and pelleted 400x $g_{av}$, 60 s and then resuspended in 1 ml 1x PBS, 37°C and spun 1000x $g_{av}$, 30 s. The washed pellet was lysed in 50 μl of 1.5x sample buffer and diluted 1:3 with 1x samples buffer and 5 μl loaded for western blotting. Time was then started following addition of either 100 μg/ml cycloheximide or an equal volume of DMSO to the remaining wells. Samples were harvested, as above, at the remaining timepoints.

## Time lapse imaging for CCNB1-GFP

Live cell imaging was carried out in 35 mm dishes with a 14 mm No. 1.5 thickness coverglass window (MatTek Corporation). HeLa CCNB1-GFP cells were seeded at 50,000 cells/imaging dish and grown for 48 hr before being treated with 2 mM thymidine for 18 hr. The thymidine was removed by washing twice with 2 ml 1x PBS and twice with 2 ml DMEM, both pre-equilibrated at 37°C, 5% $CO_2$, and imaging was started 9 hr later. Cells were imaged in a chamber maintained at 37°C, 5% $CO_2$, using an Olympus Ix81 inverted microscope, 60 × 1.42 NA oil immersion objective and an Ultraview Vox

spinning disk confocal system (Perkin Elmer). A C9100-EM-CCD camera was used for image capture. Volocity software was used for experimental set up, image projection and exporting. Cells were excited with 488 nM lasers with 100 ms exposures at 4% laser power. Cycloheximide and nocodazole were diluted in 100 µl of pre-equilibrated media before being added dropwise when necessary during the experiment, to a final concentration 100 µg/ml and 100 ng/ml, respectively.

### Biochemical and image data analysis and statistical methods

Densitometric quantification of western blots was also performed in ImageJ. Intensity was measured in a $60 \times 30$ px region of interest around each band. An adjacent region was also measured to account for background intensity. Microscopy images were processed in ImageJ. For the CCNB1-GFP fluorescence measurements over time, raw image data were opened in ImageJ using the Bio-Formats plug-in. Images were then Z-projected to sum the intensity across the volume of the whole cell. A region of interest was drawn around each cell in interphase, this shape was copied to a vacant region of the field of view for a measurement of background signal. The integrated density was then measured in these regions over time and the background subtracted. The corrected values were plotted as raw intensity. Graphs and statistical analysis were performed in GraphPad Prism. All figure layouts were created using Adobe Photoshop and Illustrator (Adobe CC).

## Acknowledgements

We thank Ulrike Gruneberg and Elena Poser for discussion and comments on the manuscript. JH was supported by a Wellcome Trust PhD award and a Cancer Research UK program grant award (C20079/A15940) supported the work of FAB.

## Additional information

### Funding

| Funder | Grant reference number | Author |
| --- | --- | --- |
| Cancer Research UK | Program Grant | James Holder<br>Francis A Barr |
| Wellcome | PhD Award | James Holder |

The funders had no role in study design, data collection and interpretation, or the decision to submit the work for publication.

### Author contributions

James Holder, Formal analysis, Investigation, Visualization, Writing - original draft, Writing - review and editing; Shabaz Mohammed, Resources, Supervision, Methodology; Francis A Barr, Conceptualization, Supervision, Funding acquisition, Writing - original draft, Writing - review and editing

### Author ORCIDs

James Holder https://orcid.org/0000-0001-7597-3104
Francis A Barr https://orcid.org/0000-0001-7518-253X

### Decision letter and Author response

Decision letter https://doi.org/10.7554/eLife.59885.sa1
Author response https://doi.org/10.7554/eLife.59885.sa2

## Additional files

### Supplementary files

• Transparent reporting form

## Data availability

Source data files have been provided for Figures 2, 3, 6 and 7. These include a processed form of the raw data, where some extraneous metadata has been removed. Full raw data is available at PRIDE with the following accession numbers: Figure 2 Total Proteome PXD019791, Figure 2 Phospho-proteome PXD019788, Figure 3 PXD019795, Figure 6 PXD019787, Figure 7 PXD019786.

The following datasets were generated:

| Author(s) | Year | Dataset title | Dataset URL | Database and Identifier |
|---|---|---|---|---|
| Holder J, Mohammed S, Barr F | 2020 | PP1 depletion phospho-proteome - Ordered dephosphorylation initiated by the selective proteolysis of cyclin B drives mitotic exit | http://proteomecentral. proteomexchange.org/ cgi/GetDataset?ID= PXD019786 | ProteomeXchange, PXD019786 |
| Holder J, Mohammed S, Barr F | 2020 | APC/C inhibition phospho-proteome - Ordered dephosphorylation initiated by the selective proteolysis of cyclin B drives mitotic exit | http://proteomecentral. proteomexchange.org/ cgi/GetDataset?ID= PXD019787 | ProteomeXchange, PXD019787 |
| Holder J, Mohammed S, Barr F | 2020 | Control phospho-proteome - Ordered dephosphorylation initiated by the selective proteolysis of cyclin B drives mitotic exit | http://proteomecentral. proteomexchange.org/ cgi/GetDataset?ID= PXD019788 | ProteomeXchange, PXD019788 |
| Holder J, Mohammed S, Barr F | 2020 | Control total-proteome - Ordered dephosphorylation initiated by the selective proteolysis of cyclin B drives mitotic exit | http://proteomecentral. proteomexchange.org/ cgi/GetDataset?ID= PXD019791 | ProteomeXchange, PXD019791 |
| Holder J, Mohammed S, Barr F | 2020 | Fractionated Control total-proteome - Ordered dephosphorylation initiated by the selective proteolysis of cyclin B drives mitotic exithibition phospho-proteome - Ordered dephosphorylation initiated by the selective proteolysis of cyclin B drives mitotic exit | http://proteomecentral. proteomexchange.org/ cgi/GetDataset?ID= PXD019795 | ProteomeXchange, PXD019795 |

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
