## [Decision Letter]

**Acceptance summary:**

This paper provides key insights into the cell cycle controls that direct exit from mitosis. Through a high-resolution proteomics approach, the authors determine the relative contributions of proteolysis and protein dephosphorylation to mitotic exit. The findings show that cyclin B destruction initiates sequential waves of dephosphorylation, dependent on the phosphatase PP1.

**Decision letter after peer review:**

Thank you for submitting your article "Ordered dephosphorylation initiated by the selective proteolysis of cyclin B drives mitotic exit" for consideration by *eLife*. Your article has been reviewed by Anna Akhmanova as the Senior Editor, a Reviewing Editor, and three reviewers. The following individuals involved in review of your submission have agreed to reveal their identity: Paola Vagnarelli (Reviewer #1); Helder Maiato (Reviewer #3).

The reviewers have discussed the reviews with one another and the Reviewing Editor has drafted this decision to help you prepare a revised submission.

Summary:

The Authors have addressed in this paper the important question of how mitotic progression is temporally regulated. The contribution of proteolysis and protein de-phosphorylation are well accepted as main mechanisms driving the exit from mitosis. However, the relative contribution and the temporal regulation are still in need of clarification. Here, the authors have set up a system to analyse and compare both protein degradation and protein de-phopshoryation in a relatively short timeline upon induction of exit form mitosis. They have presented interesting datasets that allowed the author to map the proteome during mitotic exit and the de-phosphorylation kinetics of several proteins during this stage of the cell cycle. Based on these analyses and on several manipulations of the system which included inhibiting protein degradation / protein synthesis or protein de-phosphoryaltion, they concluded that mitotic exit is temporally driven mainly by de-phosphorylations that are triggered by CDK1 inactivation and that proteolysis (apart for cyclin B) is not the main contributor to mitotic exit. This study is important and provides some provocative ideas on how exit from mitosis is regulated. This paper will be of exceptional interest after addressing a few points.

Essential revisions:

1) More information about the stages of populations of cells were analysed should be provided. With the presented data it is not clear how mitotic/cell cycle stage is inferred from the time course. The time points are very close to each other and having an idea of how many cells are really going through the process in a synchronous matter is very important. This information is missing in the paper. For example, the labels at the top of Figure 1 indicate timepoints that reflect anaphase and telophase. Time-lapse microscopy of GFP-cyclin B1 in HeLa cells showed GFP-cyclin B1 degradation starts at the metaphase-anaphase transition, but the labels in Figure 1 indicate different timing. Similarly, at the top of Figure 3B, "G1" is stated and it is not clear what this means. The timing of the forced mitotic exit system used here may not be comparable with an unperturbed mitosis. It would be useful to know, using a live cell imaging approach, how the population behaves after releasing the cell from nocodazole and then from the addition of the CDK1 inhibitor. This would also allow comparing the time course of mitotic exit to the ones for cells that are normally cycling. It has been reported that there are differences and some of them could explain the "noise" of a few datasets. A phenotype-proteome correlation could help to clarify.

2) Some statements are too strong and should be rephrased or discussed more critically.

One example is the experiments in Figure 4:

Subsection “Protein synthesis is not required for progression through the metaphase-anaphase transition”: “…There was no statistically significant difference in the mean time spent in mitosis following addition of CHX (Figure 4C). This suggests that neither mitotic progression nor SAC function is perturbed in the absence of protein synthesis”. The number of cells analysed for the experiment in 4C is quite small for the treated ones (n=7), this explains why the statistic is not strong enough. If we look at the distribution, the majority of the cells (n=5) show actually a shorter NEB-Anaphase timing. This could indicate that there is a difference. To make this statement, more cells need to be analysed or the statement needs to be removed.

3) Similarly, in Subsection “Protein synthesis is not required for progression through the metaphase-anaphase transition” the statement " the CCNB1-GFP cells treated with CHX did not display any apparent structural abnormalities during this time window (Figure 4A)" does not represent what the image shows. Just as gross analysis, it is quite visible that the cell does not properly round up, the spindle is not really well formed/shaped. This is just looking at the DIC and CCNB1-GFP, but no other characterisation has been provided. Therefore, as it stands, this conclusion is not supported by the evidence provided and more analyses are required to really demonstrate that mitotic progression does not require protein synthesis.

4) Some western blots on the kinetics and relative quantifications are based on n=1 or 2 experiment (Figure 4, Figure 5). A repeat would be appropriate since there is quite some noise in the set up as shown by the error bars in Figure 4F.

5) The experiment from Figure 7 needs some adjustments because the method to inactivate PP1 could cause indirect effects, accounting for the observations. siRNA for the catalytic subunit (more than 100 holoenzymes affected) for a few days is expected to cause a lot of defects in the phospho-proteome. If this experiment is to be included, the data should be re-evaluated. The authors have calculated the level of phosphorylation based on a normalisation to the t_0_ of the experiment. This is fine in a control situation. However, it is well described that repression of phosphatases in mitosis causes hyper-phosphorylation of the substrates. If a protein accumulates as phosphorylated in double the amount, then of course, regardless of the phosphatase, it would take double the time (or more time) in order to be de-phosphorylated. So, the apparent delayed kinetic is not a reflection of the reduced de-phosphorylation but just of the extra number of molecules that need to be processed. For example, if normally there are 10 phospho-molecules to get to 50% the cell needs to process 5 and if the capacity of the cell is 1 molecule/minute it will take 5 minutes. The time will change enormously if the starting point is 100 molecules. The prolonged time occurs even if the capacity of the cell does not change.

It is therefore important to compare the level of phosphorylation for each protein in the control and siRNA samples at t_0_ first. This would provide a normalisation baseline and allow the authors to distinguish between a problem of delayed de-phosphorylation in mitotic-exit or a problem of early mitosis hyperphoshorylation.

6) The Abstract concludes with the following statement, "selective proteolysis of cyclin B[1] drives the bulk of changes observed during mitotic exit". Treatment with flavopiridol will inhibit all CDK complexes, not just cyclin B1-CDK1. The data clearly suggest that one of the major roles of the APC/C is to reduce CDK activity. But the data do not exclude phosphorylations (possibly key ones) mediated by other cyclin-CDK complexes during mitosis that are also inhibited by flavopiridol. Do cyclin B2 and cyclin A2 levels decrease during the time course? This point should be discussed.

[Editors' note: further revisions were suggested prior to acceptance, as described below.]

Thank you for resubmitting your work entitled "Ordered dephosphorylation initiated by the selective proteolysis of cyclin B drives mitotic exit" for further consideration by *eLife*. Your revised article has been evaluated by Anna Akhmanova (Senior Editor) and a Reviewing Editor.

The manuscript has been improved but there are some remaining issues that need to be addressed before acceptance, as outlined below:

Overall, the reviewers are satisfied with the responses to their comments, however, there is one exception which relates to the re-analysis of the phosphoproteome in Figure 7, which we request is further documented. The point under discussion is the fold change in phosphorylation upon PP1 depletion. While you report in the rebuttal letter that on average the difference is 1.4-fold and argue that this level of fold-change is not major, others might have a different interpretation. Depending on the distribution of fold-changes and how the average is calculated (mean, median), a 1.4-fold global change (across all detected phosphopeptides) could be viewed as major. This average value could reflect a massive effect on a subset of proteins (including, for example, key mitotic regulators) and very little impact on other, non-PP1-regulated proteins, which when averaged together lead to a "modest" average global fold change of 1.4. To address this, it would be more informative to show a distribution of phosphorylation fold changes for all detected phosphorylation sites, potentially highlighting specific substrates or sites that show very high fold change. Readers then can make their own judgment about if this change is major or not. This is important because there are examples in the literature of substrates that change drastically (e.g. depletion of Repo-Man causes a 4 fold increase in H3-T3 phosphorylation in mitosis; Qian et al., 2010 Figure 6B).

---

## [Author Response]

Essential revisions:1) More information about the stages of populations of cells were analysed should be provided. With the presented data it is not clear how mitotic/cell cycle stage is inferred from the timecourse. The time points are very close to each other and having an idea of how many cells are really going through the process in a synchronous matter is very important. This information is missing in the paper. For example, the labels at the top of Figure 1 indicate timepoints that reflect anaphase and telophase. Timelapse microscopy of GFP-cyclin B1 in HeLa cells showed GFP-cyclin B1 degradation starts at the metaphase-anaphase transition, but the labels in Figure 1 indicate different timing. Similarly, at the top of Figure 3B, "G1" is stated and it is not clear what this means. The timing of the forced mitotic exit system used here may not be comparable with an unperturbed mitosis. It would be useful to know, using a live cell imaging approach, how the population behaves after releasing the cell from nocodazole and then from the addition of the CDK1 inhibitor. This would also allow comparing the time course of mitotic exit to the ones for cells that are normally cycling. It has been reported that there are differences and some of them could explain the "noise" of a few datasets. A phenotype-proteome correlation could help to clarify.

To answer this question, we first need to explain our methodology and the way we have assigned the different stages of mitotic exit using specific biochemical markers. This approach allows us to more accurately compare single cell data, for example of GFP-cyclin B levels with the ensemble biochemical data obtained using populations of millions of cells. To be able to make such a comparison we first need to produce highly synchronous anaphase on a scale large enough to perform biochemical experiments. The methodology uses a mitotic arrest to create a defined starting cell population, collection of mitotic cells by shake-off and then triggering synchronous anaphase entry using chemical inhibition of either CDK1-cyclin B or the checkpoint kinase MPS1 is well-established within the lab and has been published previously (Cundell et al., 2013 and 2016; Hayward et al., 2019; Serena et al., 2020). That work has used a combination of live cell imaging and biochemistry to establish thresholds for CDK1-cyclin B and PP2A-B55 phosphatase activities and shown timing of events is comparable in this system. Additionally, this experimental strategy has been used within the wider literature and shown to be comparable to normal anaphase (McCloy et al., 2015; Rogers et al., 2016). Our work builds on those studies, and to repeat that here is not necessary. We should note that it is inevitable that the timing comparisons become less exact at later time points due to the heterogeneity in cell behaviour and biochemical data.

To assign the specific stages of mitotic exit in the biochemical analysis shown in Figure 1, we marked anaphase as the point at which the anaphase-specific phosphorylation, PRC1-pT602, begins to accumulate to detectable levels. PRC1 pT602 phosphorylation is prevented during mitosis by the CDK1-dependent phosphorylation of PRC1 at T481 which is removed by PP2A-B55 in anaphase (Neef et al., 2007; Cundell et al., 2013 and 2016). We don’t differentiate anaphase A and B in this case, since sister chromatid separation can’t be easily followed in our biochemical assays. Telophase was marked as the point at which the anaphase PRC1-pT602 signal began to decline and CCNB1 signal was no longer detectable. These events are indicative of separate biochemical states defined by levels of kinase activity. In anaphase cyclin B levels are still rapidly falling, whereas in telophase they have reached a low level. By comparison to the loss of global CDK1-cyclin B activity, the activity of Aurora and PLK1 persists in anaphase and the proteolytic inactivation of these accessory kinases commences in telophase. These profiles of mitotic kinase activity and stability are critical hallmarks of directionality during mitotic exit and seemed appropriate to use as both indicators of our experimental system reproducing the expected behaviour and of approximate cell staging to aid interpretation of the results.

With respect to destruction of CCNB1, we do not claim that destruction of CCNB1 begins after anaphase onset. The relative timing of anaphase dephosphorylations and metaphase proteolytic destruction of CCNB1, securin and GMNN are altered as a direct consequence of our experimental set up. Importantly, the use of CDK1 inhibition to drive cells in mitosis uncouples CDK activity from CCNB1 level, which thus follows APC/C activity as we have modelled previously (Cundell et al., 2013). This changes the order of events, allowing early anaphase dephosphorylations to precede destruction of CCNB1.

Subsection “Ordered dephosphorylation, not proteolysis, is the dominant mode of anaphase regulation”: “Of all the detected proteins only CCNB1 and securin (Western blot only) exhibited rapid enough destruction to impact the events of anaphase. Importantly, in unperturbed cells, these degradation events occur prior to the onset of anaphase.”

To increase clarity regarding this matter we have now altered the text describing the interpretation of Figure 1.

Subsection “Ordered dephosphorylation, not proteolysis, is the dominant mode of anaphase regulation”: “The use of chemical CDK inhibition to trigger mitotic exit uncouples CDK1 activity from the level of CCNB1, allowing cells to exit mitosis prematurely prior to CCNB1 proteolysis. Consequently, some dephosphorylations take place earlier than under physiological conditions (Cundell et al., 2013; Cundell et al., 2016)”.

The labels used in Figure 3B illustrate the large difference in proteolytic rate of early and late APC/C substrates by drawing an approximate correlation between experimentally derived protein half-lives and progression through mitotic exit. They are not a definitive assignment of the cell cycle stage during which protein levels reach their minimum value. Indeed, whilst many of these proteins were stable during our window of observation, we cannot exclude accelerated rates of proteolysis occurring beyond the timescale of our analysis, leading to an over-estimation of half-life. However, we believe these qualitative assignments are appropriate, based on the wealth of published live cell data of cells exiting mitosis. In particular, 15-20 minutes after anaphase onset occurs nuclear pores have reassembled and nuclear import has restarted (Cundell et al., 2016). Furthermore, Cep55 begins to replace PLK1 at the midbody (Bastos et al., 2010) and the bulk of PRC1-pT602 signal (Figure 1) is lost 60 minutes following anaphase entry and CDK inhibition, respectively. Finally, by two hours, chromosomes have decondensed, cells have flattened and re-adhered.

2) Some statements are too strong and should be rephrased or discussed more critically.One example is the experiments in Figure 4:Subsection “Protein synthesis is not required for progression through the metaphase-anaphase transition”: “…There was no statistically significant difference in the mean time spent in mitosis following addition of CHX (Figure 4C). This suggests that neither mitotic progression nor SAC function is perturbed in the absence of protein synthesis”. The number of cells analysed for the experiment in 4C is quite small for the treated ones (n=7), this explains why the statistic is not strong enough. If we look at the distribution, the majority of the cells (n=5) show actually a shorter NEB-Anaphase timing. This could indicate that there is a difference. To make this statement, more cells need to be analysed or the statement needs to be removed.

We have revised the text and rephrased these statements. We accept that the sample size for the analysis of time spent in mitosis following blockade of protein synthesis with cycloheximide (CHX) is small. This is a consequence of the technically challenging experimental setup where we live image single cells and add CHX. If CHX is added too early, then cells fail to produce enough cyclin B to enter mitosis. It is extremely difficult to predict which late G2 cells have synthesized sufficient CCNB1 for mitotic entry prior to addition of CHX.

Subsection “Protein synthesis is not required for progression through the metaphase-anaphase transition”: “Cells which had accumulated adequate levels of CCNB1 before the addition of CHX entered mitosis (Figure 4B, lower panel, Mitotic), and spent a similar time in mitosis to the control cells (Figure 4C). […] Importantly, unlike MAD2 depletion (Michel et al., 2004), cells treated with CHX did not collapse out of mitosis, suggesting that cells were able to support spindle checkpoint function and enter anaphase following destruction of CCNB1.”

3) Similarly, in Subsection “Protein synthesis is not required for progression through the metaphase-anaphase transition” the statement " the CCNB1-GFP cells treated with CHX did not display any apparent structural abnormalities during this time window (Figure 4A)" does not represent what the image shows. Just as gross analysis, it is quite visible that the cell does not properly round up, the spindle is not really well formed/shaped. This is just looking at the DIC and CCNB1-GFP, but no other characterisation has been provided. Therefore, as it stands, this conclusion is not supported by the evidence provided and more analyses are required to really demonstrate that mitotic progression does not require protein synthesis.

We have removed the statement in question. The revised text now describes the observations more simply.

Subsection “Protein synthesis is not required for progression through the metaphase-anaphase transition”: “We then asked if protein synthesis was required to complete anaphase A and initiate cytokinesis using time lapse imaging. […] Destruction of proteins during mitotic exit was then followed by Western blotting and was found to be unaffected by inhibition of protein synthesis (Figure 4D).”

4) Some western blots on the kinetics and relative quantifications are based on n=1 or 2 experiment (Figure 4, Figure 5). A repeat would be appropriate since there is quite some noise in the set up as shown by the error bars in Figure 4F.

We have now added additional data to Figure 4D-4F and included this in the quantification so that n=3. We have not included an additional repeat of Figure 5, since Figure 6—figure supplement 1 already repeats the two main conditions from this experiment (DMSO and APC/C inhibition) using a higher rate of sampling. The quantification of the relevant timepoints has now been included in Figure 5 to more fully address this point.

5) The experiment from Figure 7 needs some adjustments because the method to inactivate PP1 could cause indirect effects, accounting for the observations. siRNA for the catalytic subunit (more than 100 holoenzymes affected) for a few days is expected to cause a lot of defects in the phospho-proteome. If this experiment is to be included, the data should be re-evaluated. The authors have calculated the level of phosphorylation based on a normalisation to the t_0_ of the experiment. This is fine in a control situation. However, it is well described that repression of phosphatases in mitosis causes hyper-phosphorylation of the substrates. If a protein accumulates as phosphorylated in double the amount, then of course, regardless of the phosphatase, it would take double the time (or more time) in order to be de-phosphorylated. So, the apparent delayed kinetic is not a reflection of the reduced de-phosphorylation but just of the extra number of molecules that need to be processed. For example, if normally there are 10 phospho-molecules to get to 50% the cell needs to process 5 and if the capacity of the cell is 1 molecule/minute it will take 5 minutes. The time will change enormously if the starting point is 100 molecules. The prolonged time occurs even if the capacity of the cell does not change.It is therefore important to compare the level of phosphorylation for each protein in the control and siRNA samples at t_0_ first. This would provide a normalisation baseline and allow the authors to distinguish between a problem of delayed de-phosphorylation in mitotic-exit or a problem of early mitosis hyperphoshorylation.

Analysis of shifts in the steady state levels of phosphorylation for individual peptides in the mitotic phospho-proteomes following depletion of catalytic subunits is possible using our methodology because of the crosswise mixing approach we use. We have now provided a revised analysis of the data set in Figure 7 which should address the concerns. To do that we used data from cross-mixed samples containing equal quantities of phospho-peptides from equivalent t_0_ samples of different conditions, e.g. t_0_ heavy siControl mixed with t_0_ hight siPPP1CA/C. These values were used to adjust the observed H/L phospho-peptide ratios from the siPPP1CA/C condition relative to control. The average ratio of total protein from siCon/siPPP1CAC cross-mixed samples, which was not the expected ratio of 1, was used to account for a technical error in quantifying amounts of protein between different conditions. Following cross-mixing, the average t_0_ phospho-peptide ratio was 1.01 and 1.42 in the control and PP1 depleted conditions, respectively. This modest increase in phosphorylation state is inconsistent with the large increase in phosphorylation occupancy required to lead to delayed dephosphorylation in the way described by the reviewers. The majority of mitotic phosphorylations are CDK-dependent and are typically high occupancy, due to the maximal CDK activity during mitosis. It has been shown that more than half of the mitotic phospho-proteome reaches >70% occupancy (Olsen et al., 2010). This reduces the potential for hyper-phosphorylation to develop for CDK substrates in the absence of opposing phosphatase activity.

To explain this data, one has to consider the way PP1 and PP2A-B55 are regulated. Because both PP2A-B55 and PP1 are inhibited during mitosis, in a CDK1-dependent manner, there is minimal global activity of these phosphatases. Consequently, although local pools of PP1 exist, inactivation of these phosphatases has little effect on the global mitotic phospho-proteome. It therefore follows that our global analysis is unaffected by the point raised in the reviews. Inactivation of CDK-opposing phosphatases will have much larger effects on the dynamics of dephosphorylation in anaphase, and potentially the G2/M transition, but not the level of steady-state mitotic phosphorylation. This was previously observed when global levels of mitotic phosphorylation did not increase following depletion of PP2A-B55 (Cundell et al., 2016). The scenario described by the reviewers may be more applicable when the mechanisms required for inhibition of PP1 and PP2A-B55 have been removed, namely PP1-T320 mutants or B55-inhibitors ENSA/ARPP19 and MASTL. Consequently, CDK would have to continually counteract the activity of PP1 and PP2A-B55, potentially shifting steady-state phosphorylation levels slightly downward. Additionally, PP2A-B56 and PP6 are active throughout mitosis and dynamically oppose locally regulated Aurora kinase activities. Therefore, inactivation of PP2A-B56 and PP6 would likely result in an upshift in steady-state phosphorylation because these substrate phosphorylation occupancies are normally limited by activity of the respective PPP.

6) The Abstract concludes with the following statement, "selective proteolysis of cyclin B[1] drives the bulk of changes observed during mitotic exit". Treatment with flavopiridol will inhibit all CDK complexes, not just cyclin B1-CDK1. The data clearly suggest that one of the major roles of the APC/C is to reduce CDK activity. But the data do not exclude phosphorylations (possibly key ones) mediated by other cyclin-CDK complexes during mitosis that are also inhibited by flavopiridol. Do cyclin B2 and cyclin A2 levels decrease during the timecourse? This point should be discussed.

Our work has to be viewed alongside the large body of published work showing that Cyclin B1 is the only major cyclin remaining immediately prior to the onset of anaphase and is the major cyclin needed to sustain mitosis during a spindle checkpoint arrest. Cyclin A2 (CCNA2) is rapidly destroyed following nuclear envelope breakdown, independent of active checkpoint signalling (Di Fiore et al., 2015; Geley et al., 2001; Jacobs et al., 2001; Wolthuis et al., 2008; Zhang et al., 2019). We can therefore conclude that CCNA2 is unlikely to contribute to regulation of mitotic exit in this system. Our approach using a mitotic arrest thus gives a biochemical state where cyclin B persists, but cyclin A levels are very low, and mimics the conditions preceding a normal metaphase-anaphase transition. In agreement with this view, CCNA2 is barely detectable during the observed experimental window during mitotic exit (Author response image 1).

**Author response image 1. sa2fig1:** Relative levels of CCNA2 and CCNB1 following wash-out from nocodazole arrest. Mitotically arrested HeLa cells were washed out from nocodazole for 25 minutes prior to CDK inhibition with flavopiridol. Samples were harvested as indicated and analysed by Western blot, with 8 µg of total lysate loaded per lane. Film exposure times are indicated next to the corresponding label. Antibodies and dilutions used are as follows: PRC1-pT481 1:4000 (AbCam; EP1514Y/ab62366), Cyclin A2 1:2000 (AbCam; ab32498), Cyclin B1 1:5000 (Millipore; 05-373), Tubulin 1:10,000 (Sigma; T6199).

Levels of CCNB2 do decrease during the course of the experiment. However, mice lacking CCNB2 are fertile and develop normally. Furthermore, CCNB2 is expressed at lower levels than CCNB1 so may not be able to provide sufficient levels of CDK activation to compensate for all functions of CCNB1 (Brandeis et al., 1998). This latter point is supported by the inconsistent detection of CCNB2 across the time course, even in fractionated total proteome samples (Figure 2—source data 1, 2 and 3 and Figure 3—source data 1). Taking into account the prior degradation of CCNA2 and the low levels of CCNB2, we would therefore argue that degradation of solely CCNB1 is the driving factor for mitotic exit. We accept it is important to address the roll of other cyclins in this context and the following text has been added to address these topics in subsection “Ordered proteolysis of cyclins during mitosis and mitotic exit”:

“Here, we have primarily focused on CCNB1 in the context of driving mitotic exit, however both CCNA2 and CCNB1 play important roles during mitosis. […] We would therefore propose that it is the destruction of solely CCNB1 which drives mitotic exit.”

[Editors' note: further revisions were suggested prior to acceptance, as described below.]

The manuscript has been improved but there are some remaining issues that need to be addressed before acceptance, as outlined below:Overall, the reviewers are satisfied with the responses to their comments, however, there is one exception which relates to the re-analysis of the phosphoproteome in Figure 7, which we request is further documented. The point under discussion is the fold change in phosphorylation upon PP1 depletion. While you report in the rebuttal letter that on average the difference is 1.4-fold and argue that this level of fold-change is not major, others might have a different interpretation. Depending on the distribution of fold-changes and how the average is calculated (mean, median), a 1.4-fold global change (across all detected phosphopeptides) could be viewed as major. This average value could reflect a massive effect on a subset of proteins (including, for example, key mitotic regulators) and very little impact on other, non-PP1-regulated proteins, which when averaged together lead to a "modest" average global fold change of 1.4. To address this, it would be more more informative to show a distribution of phosphorylation fold changes for all detected phosphorylation sites, potentially highlighting specific substrates or sites that show very high fold change. Readers then can make their own judgment about if this change is major or not. This is important because there are examples in the literature of substrates that change drastically (e.g. depletion of Repo-Man causes a 4-fold increase in H3-T3 phosphorylation in mitosis; Qian et al., 2010 Figure 6B).

To clarify the point in question we have included an additional figure within the revised manuscript, Figure 7—figure supplement 1, with the relevant data included in Figure 7—source data 1. This extended figure shows a comparison of the starting t_0_ ratios in Control and PPP1CA/C conditions, as well as specific examples of phosphorylation sites from proteins with key mitotic functions; BUB1, INCENP and Ki-67 (Figure 7—figure supplement 1A-D).

From the distribution of starting ratios, it can be seen that the majority of phospho-sites show a modest, or no, increase in steady-state mitotic phosphorylation. We suggest that these can be separated into two categories, the first are phosphorylation sites that are not subject to PP1-dependent dephosphorylation. The second category would consist of phosphorylation sites which are dephosphorylated in a PP1-dependent manner during anaphase. Importantly, the pool of PP1 responsible for the anaphase dephosphorylation of these sites would be inhibited during mitosis, meaning the mitotic abundance of these phosphorylation sites is independent of PP1. Therefore, depletion of PP1 would not result in a large change in steady-state levels of these phosphorylations rather it would delay the kinetics of dephosphorylation during anaphase.

A sub-set of ~300 phosphorylation sites demonstrate a steady state level of phosphorylation which more than doubled following depletion of PP1 (Figure 7—figure supplement 1A, >2 fold red line). This behaviour is indicative of PP1-dependent regulation during mitosis, marking these phosphorylation sites as candidate mitotic substrates of PP1 (Figure 7—source data 1). This is consistent with published data regarding H3-pT3 highlighted by the reviewers (Qian et al., 2011 and 2015). Furthermore, for these sites it would be difficult to distinguish whether any observed delay in anaphase dephosphorylation is due to the absence of anaphase-PP1 activity or due to the presence of more molecules to dephosphorylate, as previously stated by the reviewers in response to our initial submission. However, this does not impact our global analysis of dephosphorylation in the absence of PP1 as, in order to enable efficient clustering of the data, phosphorylation sites with a starting ratio of greater than 2.5 are excluded from the clustering analysis.

To address these points the following text has been added to subsection “PP1 and PP2A-B55 share a preference for basic phospho-threonine sites”:

“Depletion of PP1 has been reported to lead to hyperphosphorylation of Histone H3-T3 in prometaphase and metaphase (Qian et al., 2011; Qian et al., 2015). […] Importantly, these PP1-dependent changes occurred on both phospho-threonine and phospho-serine.”

Our global analysis of mitotic exit in the absence of PP1 activity will provide a useful list of potential PP1 substrates. However, our methodology only provides evidence of a dependence on PP1 activity not of direct dephosphorylation by PP1, nor the holoenzymes responsible for these dephosphorylations. This information, whilst important, is beyond the scope of the question we were addressing and should instead form the basis of future studies.